# ReTimeCausal: A Consistent EM Framework for Causal Discovery in Irregular Time Series

## Abstract

This paper studies causal discovery in irregularly sampled time series—a pivotal challenge in high-stakes domains like finance, healthcare, and climate science, where missing data and inconsistent sampling frequencies distort causal mechanisms. The core challenge arises from the interdependence between missing data imputation and causal structure recovery: an error in either component can cascade into the other, ultimately distorting the inferred causal graph. Existing methods either impute first and then discover, or jointly optimize both via neural representation learning, but lack explicit mechanisms to ensure mutual consistency of imputation and structure learning. We address this challenge with ReTimeCausal, an EM-based framework that alternates between imputation and structure learning, promoting structural consistency throughout the optimization process. Our framework emphasizes theoretical consistency guarantees for structure recovery, extending classical results to settings with irregular sampling and high missingness. Through kernelized sparse regression and structural constraints, ReTimeCausal iteratively refines missing values (E-step) and causal graphs (M-step), resolving cross-frequency dependencies and missing data issues. Extensive experiments on synthetic and real-world datasets demonstrate that ReTimeCausal outperforms existing state-of-the-art methods under challenging irregular sampling and missing data conditions.

## 1 Introduction

Understanding causal relationships in time series is crucial in domains such as finance, healthcare, and climate science. Unlike correlation-based analysis, causal discovery aims to identify cause-and-effect relationships and temporal mechanisms that govern complex systems (Gong et al., 2024; Assaad et al., 2022b). For instance, in finance, it helps predict market trends and manage risks (Sokolov et al., 2025; Sadeghi et al., 2023); in healthcare, it aids in developing effective medical interventions (Srikishan & Kleinberg, 2023; Smit et al., 2023; Sanchez et al., 2022); and in climate science, it enables better climate modeling and policy-making (Runge et al., 2019a). However, most existing causal discovery methods assume regularly sampled and fully observed data (Liu et al., 2023; Pamfil et al., 2020; Sun et al., 2023), which is rarely the case in real-world scenarios.

Such assumptions hinder practical causal discovery, especially in domains where time series frequently suffer from irregular sampling and missing data (Li & Marlin, 2020; Liu et al., 2025; Liu & Constantinou, 2023). For example, as shown in Figure 1(a), in the medical field, patient data may be collected at irregular intervals due to varying visit schedules or incomplete records. A patient's vital signs might be measured hourly during a hospital stay but only weekly during follow-up visits, and some data points may be missing altogether (Chauhan et al., 2024; Xu et al., 2024). This inconsistency in data collection frequency and completeness makes it difficult to apply existing causal discovery methods that assume regularly sampled and fully observed data. Naively aligning such data to a uniform time grid or ignoring missing values can distort temporal order and causal structure, obscuring key relationships. Traditional causal discovery algorithms, which rely on the assumption of fully observed and regularly sampled sequences, can lead to incorrect identification of causal relationships. In the medical context, this could result in providing inaccurate treatment recommendations, potentially delaying effective medical interventions and worsening a patient's condition, which is absolutely unacceptable.

Existing causal discovery strategies are often inadequate when dealing with time series data that exhibit irregular sampling and missing values. For example, as shown in Figure 1(b), traditional impute-then-discover pipelines, which first fill in missing entries and then apply standard structure learning algorithms, typically overlook the causal context during the imputation phase. To illustrate, consider a scenario in the medical field where a patient's vital signs are measured hourly during a hospital stay but only weekly during follow-up visits, and some data points may be missing altogether. If we naively interpolate these missing values without considering the causal relationships, we might introduce artificial correlations. For instance, if antihypertensive medication dosage ($X_1$) is only recorded monthly, while blood pressure measurements ($X_2$) are taken weekly, simple interpolation may falsely create a direct link from medication to stroke risk score ($X_3$), bypassing the mediating influence of blood pressure. Similarly, if blood pressure ($X_2$) itself is sparsely observed, naive interpolation could lead to a spurious edge from stroke risk ($X_3$) to an unrelated variable such as LDL cholesterol ($X_4$), thereby distorting the true underlying causal pathway. Alignment-based approaches introduce additional pitfalls. As shown in Figure 1(c), these methods align all variables to a common temporal grid, typically at a coarse resolution like daily or monthly, without accounting for their native sampling frequencies. This coarse alignment can obscure the true temporal ordering of events and distort causal directionality. For example, suppose $X_2$ is recorded on January 10 and $X_3$ on January 20. If both are mapped to the same monthly time bin, they may be treated as simultaneous, which can mislead the discovery algorithm to infer a reversed causal relationship, with $X_3 \rightarrow X_2$ instead of the true $X_2 \rightarrow X_3$. Such distortions are especially problematic in irregularly sampled data, where variables often evolve at different temporal scales and even small timing discrepancies may carry significant causal information. While recent iterative frameworks like CUTS and CUTS+ attempt to jointly impute and discover causal structures under irregular sampling (Cheng et al., 2023; 2024a), they rely on recurrent neural networks. Although flexible, these black-box models produce opaque representations and thus lack interpretability.

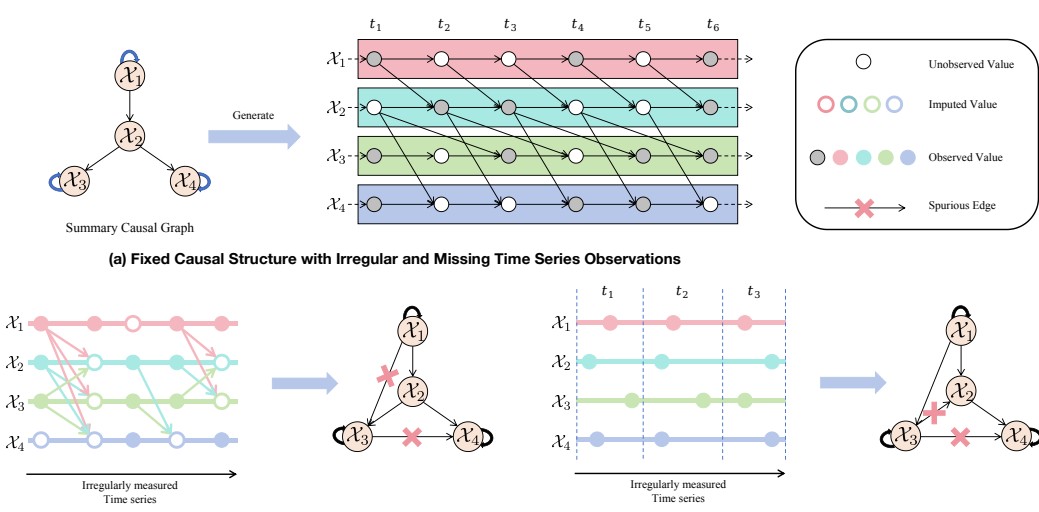

(a) Fixed Causal Structure with Irregular and Missing Time Series Observations

(b) Spurious Causal Edges Induced by Impute-Then-Discover

(c) Distorted Causality Caused by Temporal Alignment

Figure 1: (a) A true causal graph generates irregularly sampled time series with missing data. (b) Impute-then-discover methods fill in missing values without causal context (e.g., by using all observed variables at the previous timestep), which may introduce spurious edges (e.g., $X_1 \rightarrow X_3$). (c) Temporal alignment maps variables to a coarse time grid, distorting event order and potentially reversing causality ($X_3 \rightarrow X_2$).

To address the challenges of causal discovery in irregular and incomplete time series, we propose **ReTimeCausal** (**Re**covery for Irregular **Time**-series **Causal** Discovery), a framework that jointly performs missing data imputation and causal structure learning. Motivated by the intrinsic coupling between these tasks, ReTimeCausal employs an iterative refinement process in which imputations and causal graphs are updated in tandem to ensure mutual consistency.

Specifically, ReTimeCausal formulates the learning process as an Expectation-Maximization (EM) algorithm, alternating between structure-aware data imputation (E-step) and sparse causal graph

estimation (M-step). The framework accommodates both linear and nonlinear dynamics, captures lag-specific interactions, and ensures interpretability through sparsity-inducing regularization. We further provide a theoretical guarantee that the proposed joint method consistently recovers the true causal graph under given assumptions. The contributions of this work are summarized as follows:

- **A principled EM-based framework** for structure learning under severe missingness and temporal irregularity.

- **Theoretical guarantees** for causal discoveryunder irregular sampling and missing data, including consistency proofs in both idealized and realistic settings.

- **A structure-aware imputation strategy** that ensures consistency with the evolving causal graph through noise-aware completion.

- **Extensive empirical validation** on synthetic and real-world datasets shows our method consistently outperforms state-of-the-art baselines under challenging conditions (e.g., achieving an F1 score of 0.463, exceeding the best baseline's 0.414 on CausalRivers).

## 2 RELATED WORK

**Causal discovery with missing data.** Several approaches aim to identify causal structures in the presence of missing values by modeling the dependencies between variables and their missingness. Graph-based methods such as m-graphs (Bhattacharya et al., 2020; Tu et al., 2019) provide recoverability conditions under MCAR, MAR, and some MNAR settings, but are limited to static contexts and rely heavily on conditional independence testing. Additive noise model (ANM)-based methods like MissDAG (Gao et al., 2022) adopt EM-style frameworks for joint structure learning and imputation, supporting nonlinearity but still assuming fully static data. Our work builds upon this line by generalizing to irregular time series and providing consistency guarantees in dynamic settings.

**Joint imputation and causal discovery in irregular time series.** Recent methods like CUTS (Cheng et al., 2023) and CUTS+ (Cheng et al., 2024a) integrate causal discovery with imputation in irregularly sampled time series using neural architectures. While effective, these black-box models lack interpretability and theoretical transparency. Neural Graphical Modeling (Bellot et al., 2021) uses continuous-time Neural ODEs to avoid interpolation, but assumes infinitesimal-time dependencies. In contrast, our method retains interpretability by using sparse regression over discrete lagged variables, and offers theoretical guarantees under irregular sampling and high missingness. An extended discussion is provided in Appendix A.1.

## 3 PROBLEM SETUP

ReTimeCausal aims to uncover the underlying causal structure in multivariate time series data characterized by missing values and temporal irregularities. As formalized in Assumption 3.2, we assume an ignorable missingness mechanism (MCAR or MAR), consistent with prior work such as CUTS, CUTS+, and MissDAG (Cheng et al., 2023; 2024a; Gao et al., 2022). Under these assumptions, it is theoretically justified that unbiased causal inference can be achieved without explicitly modeling the missingness process. We further assume that the observed data is generated from a structural causal model with finite-order temporal dependencies, forming the foundation for our method's design and analysis.

The Additive Noise Model (ANM) is a widely adopted structural causal framework in the causal discovery literature, where each variable is expressed as a function of its causal parents plus an independent noise term (Hoyer et al., 2008; Uemura et al., 2022; Gao et al., 2022; Montagna et al., 2023). At each time step $t$, the observed variables $\mathbf{X}^t = \{X_i^t\}_{i=1}^d$ are defined as:

$$X_i^t = f_i(\mathbf{Pa}_i^t) + \epsilon_i^t \tag{1}$$

where $\mathbf{Pa}_i^t$ denotes the set of parent variables of $X_i^t$, and $\epsilon_i^t$ is an exogenous noise term, assumed independent of $\mathrm{Pa}_i^t$ and of other contemporaneous noise terms. This formulation supports both linear and nonlinear mechanisms and enables identifiability under standard conditions.

In real-world scenarios, $\mathbf{X}_o$ may correspond to vital signs collected at high frequency during a patient's hospitalization (e.g., hourly measurements of heart rate or blood pressure), while $\mathbf{X}_m$ arises

from periods with sparse or missing records, such as during post-discharge follow-ups or due to incomplete documentation. We denote a multivariate time series as $\mathcal{D} = \{\mathbf{X}_o, \mathbf{X}_m\}$.

**Definition 3.1** (Summary Causal Graph). A summary causal graph is a directed graph over the variable set $\{\mathcal{X}_1, \ldots, \mathcal{X}_d\}$. There is a directed edge from $\mathcal{X}_j$ to $\mathcal{X}_i$ whenever there exist time indices $t_1 < t_2$ such that the underlying time-indexed graph contains a direct causal arrow $X_j^{t_1} \to X_i^{t_2}$. Self-loops are allowed when a variable influences its own future values.

**Assumption 3.2** (Ignorable Missingness). The missingness mechanism is said to be ignorable if it falls under either the Missing Completely at Random (MCAR) or Missing at Random (MAR) conditions. In such cases, the probability that an entry is missing depends only on the observed data, and not on the unobserved (missing) values themselves.

**Assumption 3.3** (Finite-Order Markov Property). The time series follows a finite-order Markov process of order $L$, in which the state at time $t+1$ depends only on the most recent $L$ time steps:
$$P(\mathbf{X}^{t+1} \mid \mathbf{X}^t, \mathbf{X}^{t-1}, ..., \mathbf{X}^1) = P(\mathbf{X}^{t+1} \mid \mathbf{X}^t, ..., \mathbf{X}^{t-L+1}).$$

**Assumption 3.4** (Consistency Throughout Time, Definition 7 in Assaad et al. (2022b)). A causal graph $\mathcal{G}_S$ for a multivariate time series $\mathbf{X}$ is said to be consistent throughout time if all the causal relationships remain constant in direction throughout time.

**Assumption 3.5** (Causal Sufficiency). A set of variables is said to be causally sufficient if all common causes of all variables are observed:
$$\forall \mathcal{X}_i, \quad \text{if } \exists Z \text{ such that } Z \to \mathcal{X}_i, \text{ then } Z \in \{\mathcal{X}_1, ..., \mathcal{X}_d\}.$$

**Assumption 3.6** (Faithfulness). The data-generating process is faithful to the true summary causal graph. For any two variables $\mathcal{X}_i$ and $\mathcal{X}_j$, and any subset $S \subseteq \{\mathcal{X}_1, \ldots, \mathcal{X}_d\} \setminus \{\mathcal{X}_i, \mathcal{X}_j\}$, If $\mathcal{X}_i \not\perp\!\!\!\perp \mathcal{X}_j \mid S$, then $\mathcal{X}_i$ and $\mathcal{X}_j$ are connected in $\mathcal{G}_S$ given $S$.

For completeness, a consolidated list of all notations is provided in Appendix A.2.

The ANM provides a well-established foundation for causal discovery in fully observed data, capturing both linear and nonlinear dependencies. However, existing theoretical results primarily address settings with complete and regularly sampled observations. While some recent work has studied causal discovery with missing values under specific constraints in static settings (Gao et al., 2022; Qiao et al., 2024), theoretical guarantees for time series data with irregular sampling remain limited. Under our modeling assumptions, we provide a formal guarantee of correctness in this more general setting: Proposition 1 establishes that our EM-style joint optimization framework recovers the true causal structure in the asymptotic regime.

## 4 ReTimeCausal

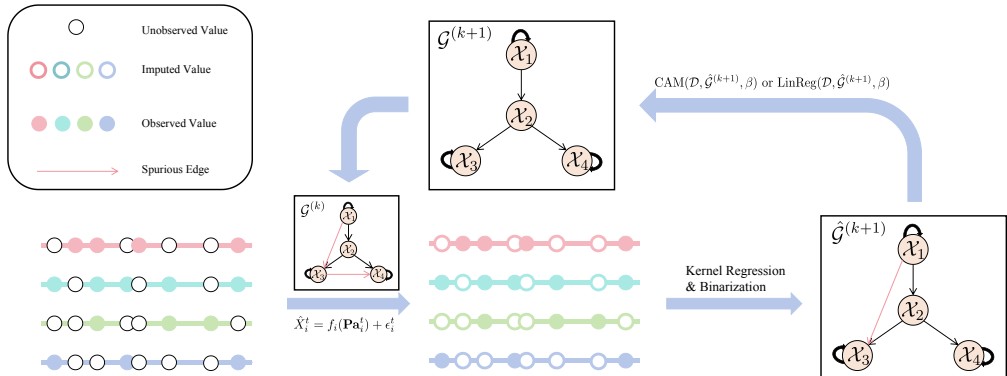

Figure 2: Overview of the ReTimeCausal Framework

Motivated by the limitations of existing methods in handling missing and irregular time series, we propose **ReTimeCausal**, a framework that integrates Additive Noise Models (ANMs) with an EM-

style optimization to jointly perform imputation and causal discovery. The goal is to recover sparse, interpretable, and lagged causal graphs without relying on black-box models.

As shown in Figure 2, ReTimeCausal consists of three components: (1) an *imputation module* that estimates missing values based on current structural functions; (2) a *regression module* that recovers causal coefficient matrices via sparse regression, followed by thresholding; and (3) a *pruning mechanism* that removes spurious edges. These steps iterate via an EM-style loop, progressively refining both data and structure to align with the underlying causal process.

## 4.1 EM-BASED IMPUTATION

Unlike impute-then-discover pipelines, our EM-based design ensures structure-aware imputation. This joint treatment of imputation and structure learning is realized via an Expectation-Maximization method, where each iteration consists of two alternating steps:

First, in the **E-step**, given the current parameters $\theta^{(k)}$ and the estimated summary causal graph $\mathcal{G}_S^{(k)}$, we compute the expected complete-data log-likelihood:

$$Q(\theta, \theta^{(k)}, \mathcal{G}_S^{(k)}) = \mathbb{E}_{\mathbf{X}_m | \mathbf{X}_o; \theta^{(k)}} \left[ \log p(\mathbf{X}_o, \mathbf{X}_m \mid \theta^{(k)}, \mathcal{G}_S^{(k)}) \right] \tag{2}$$

Missing values $X_i^t$ are imputed via their conditional expectations:

$$\hat{X}_i^t = \mathbb{E}[X_i^t \mid \mathbf{Pa}_i^t, X_{1:d}^{t-L:t-1}; \theta^{(k)}] \tag{3}$$

Under the ANM with zero-mean noise $\mathbb{E}[\epsilon_i] = 0$, the conditional expectation of each variable directly reduces to its corresponding structural function:

$$\hat{X}_i^t = f_i(\mathbf{Pa}_i^t) \tag{4}$$

Second, in the **M-step**, the parameters of the structural functions are optimized by maximizing the expected complete-data log-likelihood under the current graph estimate:

$$\theta^{(k+1)} = \arg\max_\theta Q(\theta, \theta^{(k)}, \mathcal{G}_S^{(k)})$$

The causal graph is subsequently estimated from the imputed data:

$$\mathcal{G}_S^{(k+1)} = \mathcal{S}\left(\hat{\mathbf{X}}^{(k)}\right)$$

where $\mathcal{S}(\cdot)$ denotes the structure learning function, realized by regression and further refined through binarization and pruning in accordance with the assumed causal mechanism.

## 4.2 KERNELIZED SPARSE REGRESSION FOR NONLINEAR CAUSAL DISCOVERY

To fully utilize the potential of EM in nonlinear causal settings, we integrate kernel-based sparse regression into the M-step. This section presents the theoretical rationale for this integration and introduces kernel regression as an effective tool for modeling complex dependencies.

### 4.2.1 EM-BASED ANM

To capture nonlinear causal dependencies within the EM framework introduced in Section 4.1, we adopt kernel-based regression, which enables universal function approximation (Schölkopf & Smola, 2002) and preserves ANM identifiability under non-Gaussian noise (Hoyer et al., 2008).

Formally, each input vector $\mathbf{X}^t \in \mathbb{R}^d$ is mapped into a high-dimensional feature space via a kernel embedding $\phi : \mathbb{R}^d \to \mathcal{H}$, where $\mathcal{H}$ denotes an RKHS, and inner products $\langle \phi(x), \phi(x') \rangle$ are computed using a Mercer kernel $\kappa(x, x')$. For computational tractability, we approximate this mapping using a finite-dimensional representation $\phi(\mathbf{X}^t) \in \mathbb{R}^p$, where $p$ is the number of kernel features determined by the approximation method. The temporal dynamics are modeled linearly in the feature space as:

$$\phi(\mathbf{X}^t) = \sum_{\tau=1}^{L} \mathbf{W}_{\text{high}}^{(\tau)} \phi(\mathbf{X}^{t-\tau}) + \boldsymbol{\epsilon}^t \tag{5}$$

where each $\mathbf{W}_{\text{high}}^{(\tau)} \in \mathbb{R}^{p \times d}$ is a lag-specific transformation matrix. This formulation implies that the observed variable $X_i^t$ depends nonlinearly on past multivariate lags $\mathbf{X}^{t-\tau}$, thereby supporting a broad class of nonlinear autoregressive models (NARs). The design is expressive enough to capture both short-term and long-range dependencies.

### 4.2.2 STRUCTURE RECOVERY AND SPARSITY

Since kernelized operations are inherently non-interpretable in the input space, we introduce a projection mechanism to recover input-level causal structures. Let $\Phi_{k,j}$ denote the influence of the $j$-th input dimension on the $k$-th basis in feature space. The reconstructed influence of variable $X_j^{t-\tau}$ on $X_i^t$ at lag $\tau$ is given by:

$$W_{j,i}^{(\tau)} = \sum_{r=1}^{p} \Phi_{j,r} \cdot W_{\text{high},r,i}^{(\tau)} \tag{6}$$

The collection $\{\mathbf{W}^{(\tau)}\}_{\tau=1}^{L}$ defines an interpretable, approximate summary of lagged structural causal relationships under the ANM. The projection into input space supports thresholding and pruning, improving identifiability and model sparsity.

We minimize a composite loss that balances prediction accuracy in RKHS with structural sparsity:

$$\min_{\mathbf{W}_{\text{high}}} \sum_{t=L}^{T} \left\| \phi(\mathbf{X}^t) - \sum_{\tau=1}^{L} \mathbf{W}_{\text{high}}^{(\tau)} \phi(\mathbf{X}^{t-\tau}) \right\|_2^2 + \lambda \sum_{\tau=1}^{L} \|\mathbf{W}_{\text{high}}^{(\tau)}\|_1. \tag{7}$$

The $\ell_1$ regularization on the projected weights $\mathbf{W}_{\text{high}}$ encourages a sparse causal graph, analogous to Lasso-based feature selection (Tibshirani, 1996). This allows us to differentiate spurious correlations from direct causal effects even in the presence of nonlinear dynamics.

To construct the final causal graph, we apply a thresholding operation to the recovered weights. Specifically, for a given threshold $\gamma > 0$, we define the binary adjacency $\hat{\mathcal{G}}_S^{(k,\tau)} \in \{0,1\}^{d \times d}$ as:

$$\hat{\mathcal{G}}_S^{(k,\tau)} = \mathbb{I}\left( |\mathbf{W}^{(\tau)}| > \gamma \right) \tag{8}$$

This projection-based thresholding strategy removes weak or spurious dependencies, producing a sparse and interpretable lagged causal graph in the original input space. By recovering variable-level influence weights from the kernel feature space, it enables principled post-processing such as coefficient-based pruning and significance filtering, which in turn enhance structural clarity and robustness, particularly in high-dimensional or noisy settings.

Without loss of generality, when the kernel function degenerates to the identity function or when nonlinearities are mild, the entire model formulation reduces to a standard sparse linear regression over lagged variables. In such cases, the projected weight matrices $\mathbf{W}^{(\tau)}$ directly correspond to causal coefficients in the original input space, and structure learning becomes equivalent to sparse linear Granger modeling. To illustrate this special case and enhance interpretability, we provide a linear instantiation of our method in Appendix A.3.

### 4.3 NOISE-AWARE IMPUTATION FOR PRUNING-CONSISTENT RECOVERY

Causal pruning techniques such as Causal Additive Models (CAM) (Bühlmann et al., 2013) rely on a fundamental assumption: the noise term must be statistically independent from its parent variables. However, this assumption can be violated when missing values are imputed deterministically using Eq. 4. In this case, the imputed values lie exactly on the regression manifold, effectively removing the noise term and introducing spurious dependence between $\hat{X}_i^t$ and its predictors. This undermines the post hoc pruning methods such as CAM, which may fail to identify false causal edges.

To prevent such artifacts, we propose a noise-aware imputation strategy for nonlinear models. Specifically, we modify the imputation step to explicitly inject noise during data completion:

$$\hat{X}_i^t = f_i(\mathbf{Pa}_i^t) + \epsilon_i^t \tag{9}$$

where $\epsilon_i^t$ is a noise term sampled from the empirical distribution of residuals $\hat{\epsilon}_i^t = X_i^t - \hat{f}_i(\mathbf{Pa}_i^t)$ to match the learned noise profile. This adjustment helps restore the independence assumption between $\epsilon_i^t$ and $\mathbf{Pa}_i^t$, thereby making the downstream pruning process statistically sound. Empirically, we find that this noise-injection mechanism substantially improves pruning quality when applying CAM to kernel-based nonlinear models. It enables more accurate discrimination between true and spurious causal links, particularly in high-dimensional settings with partial observability. The full pruning method, including CAM and linear-model variants, is detailed in Appendix A.4.

## 4.4 ITERATIVE OPTIMIZATION AND THEORETICAL ANALYSIS

This section provides implementation details of the underlying models, describes the iterative optimization method, and presents the theoretical guarantees that underpin ReTimeCausal.

### 4.4.1 NETWORK AND MODEL IMPLEMENTATION DETAILS

To model complex causal dependencies, each structural function $f_i(\cdot)$ is parameterized as a three-layer feedforward neural network and optimized via gradient descent during the model update step. This design follows recent trends in causal discovery that integrate neural estimation into iterative learning methods (Wu et al., 2023; Lachapelle et al., 2020; Brouillard et al., 2020; Wang et al., 2020), enabling flexible modeling of both linear and nonlinear mechanisms. For nonlinear settings, we adopt the Gaussian Radial Basis Function (RBF) kernel, defined as:$\kappa(x, x') = \exp\left(-\frac{\|x-x'\|^2}{2\sigma^2}\right)$, where $\sigma$ is a bandwidth hyperparameter. The RBF kernel is chosen for its universal approximation capabilities and smoothness properties, making it particularly well-suited for modeling nonlinear causal mechanisms under the ANM framework.

### 4.4.2 ITERATIVE OPTIMIZATION

ReTimeCausal operates as an alternating optimization method, where each iteration comprises two core phases: imputing missing values using current causal estimates, and updating model parameters and structure using the completed data. This cycle continues until convergence. To improve robustness across iterations, we apply exponential smoothing to the estimated weight matrices:

$$\mathbf{W}^{(k)} = \alpha \mathbf{W}^{(k-1)} + (1-\alpha)\hat{\mathbf{W}}^{(k)}, \tag{10}$$

where $\alpha \in [0, 1]$ is a smoothing coefficient. We then apply thresholding to derive the binary graph $\mathcal{G}_S^{(k)}$, followed by model-specific pruning.

While ReTimeCausal adopts an imputation-update structure inspired by EM, it departs from the classical formulation by using direct optimization with neural estimators, which provides greater flexibility and modularity. A summary of the algorithmic method is provided in Appendix A.5.

### 4.4.3 THEORETICAL CONVERGENCE GUARANTEES

**Proposition 1 (Structural Consistency)** *Let the variable sequence $\{\mathbf{X}^1, ..., \mathbf{X}^t\}$ be generated by a finite-order Markov process that satisfies Assumptions 3.2–3.6. If the regularization and thresholding parameters used in the algorithm decrease appropriately with the sample size, and the smoothing weight remains within a fixed range, then the estimated summary causal graph $\hat{\mathcal{G}}_S^{(k)}$ converges structurally to the true graph $\mathcal{G}_S^*$, satisfying:* $\lim_{k\to\infty} \lim_{n\to\infty} \Pr\left(\hat{\mathcal{G}}_S^{(k)} = \mathcal{G}_S^*\right) = 1.$

Proposition 1 establishes the asymptotic consistency of ReTimeCausal under standard assumptions. In particular, it suggests that, given sufficient data and a well-initialized optimization process, the algorithm can recover the true underlying causal structure in the limit. The proof builds on convergence results under missing data (Wu, 1983), consistency of universal function approximators (Schölkopf & Smola, 2002), and sparse structure recovery guarantees (Wainwright, 2009) in high-dimensional sparse regression (Wainwright, 2009). A formal proof is provided in Appendix A.7.1. Beyond the asymptotic result, we further discuss finite-sample and finite-iteration behavior and provide a derivation in the Appendix A.7.2. These guarantees rest on the ANM, which underpins both imputation and structure learning by separating true causal effects from noise, ensuring recovery of genuine dependencies over spurious patterns.

## 5 EXPERIMENTS

In this section, we evaluate the performance of ReTimeCausal on both synthetic and real-world datasets, under varying levels of missingness, graph sizes, and causal complexities. For clarity, we report only mean values in most result tables, as the standard deviations are consistently below 0.001 and do not materially affect the comparative trends.

### 5.1 BASELINE ALGORITHMS

We compare ReTimeCausal against representative methods from different paradigms: PCMCI (Runge et al., 2019a) (constraint-based), DYNOTEARS (Pamfil et al., 2020) (score-based), CUTS+ (Cheng et al., 2024a) (iterative neural framework for irregular series), Rhino (Gong et al., 2023) (deep nonlinear temporal causal model), and AERCA (Han et al., 2025) (root-cause model with Granger-based structure learning). These baselines collectively cover classical causal discovery approaches as well as recent models tailored to irregular or high-dimensional time series. Since most methods require complete data, we additionally apply TimeMixer as a preprocessing step for fair comparison (Wang et al., 2024). Detailed descriptions of each baseline and evaluation metric are provided in Appendix A.8.

### 5.2 EXPERIMENTS ON SYNTHETIC DATA

We evaluate our proposed algorithm on synthetic time series data generated from Eq.1, incorporating both linear and nonlinear causal mechanisms. The data generation strictly follows the assumptions outlined in Section 3. The causal graph is generated using the Erdős–Rényi model, and weights are uniformly scaled to ensure numerical stability. Missing values are introduced randomly under the assumption 3.2, with missing rates (MR) ranging from 20% to 80%.

Following the design in Wu et al. (2024), we simulate causal mechanisms by applying element-wise transformations to parent variables before aggregation:

$$X_i^t = \sum_{\tau=1}^{L} \sum_{j=1}^{d} W_{ij}^{(\tau)} \cdot f_i(X_j^{t-\tau}) + \epsilon_i^t \tag{11}$$

We instantiate both linear and nonlinear settings within this general framework by varying the transformation function $f_i(\cdot)$ and the noise distribution. In both cases, additive noise is drawn from either Gaussian or Laplace distributions. In the linear setting, each variable is a weighted sum of its lagged parents. In the nonlinear setting, $f_i(\cdot)$ is selected from nonlinear functions such as $\sin$ or $\tanh$, and fix the function type for each experiment. This leads to more complex functional dependencies and stronger temporal entanglement. Similarly, we use [FunctionType]-[NoiseType]-[NumVars]-[NumEdges]-[MaxLag] to describe synthetic data configurations throughout the experiments.

Table 1: Effect of Missing Data Rates on Model Performance in Synthetic Time Series

| Methods | LR-gaussian-10-10-2 | | | LR-gaussian-20-20-2 | | |
|---|---|---|---|---|---|---|
| | MR=0.0 | MR=0.6 | MR=0.8 | MR=0.0 | MR=0.6 | MR=0.8 |
| PCMCI+TimeMixer | 0.889 | 0.400 | 0.294 | 0.857 | 0.152 | 0.141 |
| PCMCI+EM | 0.889 | 0.439 | 0.375 | 0.857 | 0.205 | 0.190 |
| DYNOTEARS+TimeMixer | 1.000 | 0.263 | 0.277 | 0.976 | 0.173 | 0.155 |
| Rhino+TimeMixer | 0.286 | 0.235 | 0.162 | 0.123 | 0.243 | 0.185 |
| AERCA+TimeMixer | 0.312 | 0.225 | 0.234 | 0.098 | 0.065 | 0.083 |
| CUTS+ | 0.211 | 0.217 | 0.212 | 0.104 | 0.133 | 0.124 |
| ReTimeCausal | **1.000** | **1.000** | **1.000** | **0.976** | **0.930** | **1.000** |

As shown in Table 1, under linear-Gaussian dynamics with 10 and 20 variables, ReTimeCausal outperforms all competing approaches across all missing data rates. Specifically, in the 20-variable scenario, it achieves an F1 score of 0.976 with fully observed data, maintains a high score of 0.930 when 60% of the data is missing, and remarkably reaches a perfect score of 1.000 even under

80% missingness—demonstrating exceptional robustness. In contrast, The methods that rely on TimeMixer-imputed data show severe performance drop, with F1 scores falling below 0.2 in larger-scale settings with high missing rates. Although CUTS+ is designed for irregularly sampled data, it also fails to sustain performance under such conditions, consistently yielding F1 scores below 0.13. This highlights the limitations of methods that rely on Granger-style assumptions and weak functional modeling, particularly when data quality is compromised by substantial missingness.

Table 2 further evaluates the scalability of ReTimeCausal under increasing graph sizes, with a fixed 60% missing rate and TANH-based nonlinear causal functions. For 10 variables, ReTime-Causal achieves an F1 score of 0.909, an SHD of 4, and a low SID of 2, indicating strong alignment with both structural and interventional ground truth. As the graph expands to 20 and 50 variables, it maintains strong performance with F1 scores of 0.923 and 0.816, SHDs of 6 and 39, and SID values of 4 and 39.5. Meanwhile, all baseline methods exhibit significant performance degradation in both structural accuracy and interventional consistency. For instance, PCMCI+TimeMixer and PCMCI+EM report SID values exceeding 80 and 100 for the 20-variable setting, while CUTS+—despite reasonable SID scores in smaller graphs—reaches a SID of 235 for 50 variables, indicating instability in high-dimensional scenarios.

Table 2: Scalability Evaluation on Nonlinear Synthetic Graphs of Increasing Size

| Methods | TANH-laplace-10-20 | | | TANH-laplace-20-40 | | | TANH-laplace-50-100 | | |
|---|---|---|---|---|---|---|---|---|---|
| | SHD | SID | F1 | SHD | SID | F1 | SHD | SID | F1 |
| PCMCI+TimeMixer | 26 | 25 | 0.430 | 86 | 84 | 0.259 | 400 | 255 | 0.141 |
| PCMCI+EM | 15 | 27 | 0.545 | 43 | 110 | 0.318 | 124 | 216 | 0.311 |
| DYNOTEARS*+TimeMixer | 58 | 11 | 0.393 | 307 | 73 | 0.177 | 1717 | 84 | 0.092 |
| Rhino+TimeMixer | 43 | 15 | 0.308 | 263 | 55 | 0.165 | 253 | 258 | 0.087 |
| AERCA+TimeMixer | 52 | 22 | 0.312 | 194 | 55 | 0.192 | 1250 | 149 | 0.074 |
| CUTS+ | 70 | 3 | 0.352 | 361 | 6 | 0.178 | 1145 | 235 | 0.085 |
| ReTimeCausal | **4** | **2** | **0.909** | **6** | **4** | **0.923** | **39** | **39.5** | **0.816** |

These results demonstrate that ReTimeCausal achieves accurate and scalable causal discovery in settings with high missingness, large variable dimensions, and nonlinear interactions. Its joint EM-style optimization and kernel-based modeling strategy allows it to avoid the structural inconsistencies and performance collapse observed in conventional impute-then-discover pipelines.

## 5.3 Experiments on real-world Data

To evaluate our method on real-world data, we utilize a 10-node subgraph from the RiversEast-Germany portion of the CausalRivers benchmark (Gideon et al., 2025), which includes a known ground truth causal graph. We simulate missing values by randomly removing 20% and 60% of the time-series data.

Table 3: Empirical results (mean$_{\pm std}$) on CausalRivers Dataset

| Methods | CausalRivers (MR = 0.2) | | | CausalRivers (MR = 0.6) | | |
|---|---|---|---|---|---|---|
| | SHD | SID | F1 | SHD | SID | F1 |
| PCMCI+TimeMixer | $\mathbf{13.0}_{\pm 0.00}$ | $54.0_{\pm 0.00}$ | $0.414_{\pm 0.01}$ | $\mathbf{16.0}_{\pm 0.00}$ | $59.0_{\pm 0.00}$ | $0.273_{\pm 0.00}$ |
| PCMCI+EM | $23.3_{\pm 2.52}$ | $55.0_{\pm 2.00}$ | $0.253_{\pm 0.03}$ | $23.7_{\pm 1.53}$ | $55.3_{\pm 2.08}$ | $0.203_{\pm 0.01}$ |
| DYNOTEARS*+TimeMixer | $59.0_{\pm 1.00}$ | $43.0_{\pm 0.00}$ | $0.392_{\pm 0.00}$ | $47.3_{\pm 0.58}$ | $13.0_{\pm 0.00}$ | $0.355_{\pm 0.00}$ |
| Rhino+TimeMixer | $45.2_{\pm 12.1}$ | $34.8_{\pm 8.41}$ | $0.260_{\pm 0.03}$ | $51.4_{\pm 3.21}$ | $30.0_{\pm 9.00}$ | $0.265_{\pm 0.06}$ |
| AERCA+TimeMixer | $47.4_{\pm 1.67}$ | $\mathbf{11.8}_{\pm 3.35}$ | $0.400_{\pm 0.02}$ | $51.0_{\pm 2.00}$ | $18.0_{\pm 4.00}$ | $0.354_{\pm 0.03}$ |
| CUTS+ | $31.0_{\pm 1.00}$ | $16.0_{\pm 0.00}$ | $0.367_{\pm 0.00}$ | $38.0_{\pm 0.00}$ | $35.0_{\pm 0.58}$ | $0.319_{\pm 0.00}$ |
| ReTimeCausal | $36.3_{\pm 1.15}$ | $16.0_{\pm 2.00}$ | $\mathbf{0.463}_{\pm 0.02}$ | $48.3_{\pm 5.86}$ | $\mathbf{8.0}_{\pm 1.73}$ | $\mathbf{0.414}_{\pm 0.03}$ |

As shown in Table 3, although overall performance remains modest on this challenging real-world dataset, ReTimeCausal consistently achieves the best results across both missing rates. In particular, it attains the highest F1 scores (0.463 at 20% missingness and 0.414 at 60%), outperforming the strongest baseline by approximately 5% points in each case. Moreover, ReTimeCausal achieves the

lowest SID (8.0) at high missingness, demonstrating improved robustness to data sparsity and irregular sampling. It is important to note that PCMCI+TimeMixer shows relatively strong performance when the missing rate is 0.2 (F1 = 0.414), but this advantage does not persist when the missing rate increases to 0.6. This pattern indicates that traditional imputation-based pipelines are more likely to introduce misleading dependencies under high missingness.

## 6    CONCLUSION

In this paper, we propose ReTimeCausal, an EM-augmented additive noise modeling framework for interpretable causal discovery in the presence of irregular time series data. ReTimeCausal integrates Expectation-Maximization with additive noise models, enabling joint imputation and structure learning while preserving temporal resolution. Unlike impute-then-discover pipelines, our method learns lag-specific causal dependencies in a noise-aware and interpretable manner. Theoretical analysis and empirical results demonstrate its robustness under linear and nonlinear settings with high missingness. Future work includes (1) extending ReTimeCausal to handle MNAR data via explicit modeling of missingness mechanisms, (2) improving computational efficiency through parallelizable or variational EM techniques, and (3) generalizing the framework to support latent confounders. We discuss limitations in Appendix A.12.

## REPRODUCIBILITY STATEMENT

We have made efforts to ensure the reproducibility of our work. On the theoretical front, all core assumptions are explicitly stated in Section 3, with additional discussions provided in Appendix A.6. The main structural consistency proposition is presented in Section 4.4.3, and a formal proof is included in Appendix A.7.1. Furthermore, we derive a convergence theorem under realistic settings in Appendix A.7.2, with its full proof provided.

On the methodological side, Section 4 provides a detailed description of the ReTimeCausal framework, including the EM-based imputation strategy, kernelized sparse regression module, and the pruning mechanism. To improve interpretability and transparency, we also provide a linear variant of our method as a special case, described in Appendix A.3.

For the experimental setup, all synthetic datasets are generated based on the formulation described in Eq. 11, as referenced in Section 5.2. Implementation details of the network architecture and optimization procedure are summarized in Section 4.4.1, with the overall training and inference workflow outlined in Appendix A.5. Relevant hyperparameter settings and experimental configurations are also reported in Appendix A.11.

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

# A  APPENDIX

## A.1  RELATED WORK

**Graph-based identification methods** aim to characterize the identifiability of causal structures under missing data by explicitly modeling the dependencies between variables and their missingness indicators. Approaches such as m-graphs (Bhattacharya et al., 2020; Tu et al., 2019) provide necessary and sufficient conditions for recoverability under MCAR, MAR, and some MNAR mechanisms. These methods enhance theoretical understanding by formalizing the graphical interplay between causal and missingness structures. However, they are mainly designed for static settings and rely on conditional independence testing, which becomes unreliable when data is sparse or temporally structured.

**Additive noise model-based approaches** extend structural causal models to handle missing values via generative modeling. For example, MissDAG (Gao et al., 2022) employs EM-style optimization to jointly learn causal graphs and impute missing data under MCAR/MAR assumptions, while Qiao et al. (2024) further extend this line by establishing identifiability under weak self-masking MNAR conditions. These methods support nonlinearity and partial observability, but are primarily designed

for static data with complete conditional models. In contrast, our method extends this paradigm to irregular time series by explicitly modeling temporal dependencies through lagged causal effects and leveraging sparse kernel regression for interpretable nonlinear estimation. Moreover, we provide finite-sample and finite-iteration guarantees for structure recovery, complementing existing asymptotic identifiability results.

**Joint imputation and causal discovery methods** represent a recent advancement in handling irregular time series with missing data. Cheng et al. (2023) proposes an iterative framework that alternates between latent data prediction and sparse neural Granger graph estimation, enabling mutual refinement between imputed values and the learned causal structure. Cheng et al. (2024a) extends this paradigm to high-dimensional settings through a coarse-to-fine discovery strategy combined with message-passing GNNs, effectively addressing both feature redundancy and scalability. These models are among the few that jointly accommodate irregular sampling and high dimensionality, and consistently outperform classical pipelines that separate imputation from structure learning.

Beyond these, several recent methods aim to tackle causal discovery under irregular sampling. The Neural Graphical Modeling framework (Bellot et al., 2021) introduces a continuous-time formulation using Neural ODEs with local independence constraints. While it provides consistency guarantees and avoids interpolation, it focuses on infinitesimal-time dependencies and assumes a continuous underlying process. In contrast, our method models lagged causal interactions directly at the discrete level, providing interpretable structures and finite-sample guarantees. The entropy-based method (Assaad et al., 2022a) infers summary causal graphs without interpolation, which provides robustness to sparse sampling. However, it lacks resolution at the lag level and does not model temporal causality explicitly. Our framework bridges this gap by supporting irregular timestamps while recovering lag-specific, nonlinear causal dependencies.

## A.2 NOTATIONS USED THROUGHOUT THIS PAPER

To improve clarity and readability, we summarize the key mathematical symbols used throughout the paper in Table 4.

Table 4: List of Notations used throughout the paper.

| Symbol | Definition |
|---|---|
| $\mathbf{X}_o, \mathbf{X}_m$ | Observed and missing subsets of the time series $\mathcal{D}$ |
| $\mathbf{X}^t$ | Multivariate observation at time step $t$ |
| $\mathcal{X}_i$ | The $i$-th variable (node) in the summary causal graph |
| $\mathcal{G}_S$ | Summary causal graph over $\{\mathcal{X}_1, \ldots, \mathcal{X}_d\}$ |
| $L$ | Maximum lag order considered in temporal dependencies |
| $T$ | Length of the time series |
| $f_i(\cdot)$ | Structural function mapping from causal parents $\mathbf{Pa}_i$ to variable $X_i$ |
| $\epsilon_t$ | Exogenous noise term (independent of causal parents) |
| $\mathbf{W}^{(\tau)}$ | Weight matrix capturing causal influence at lag $\tau$ |
| $\mathbf{W}_{\text{high}}^{(\tau)}$ | High-dimensional kernel weight matrix in RKHS |
| $\Phi$ | Projection matrix from kernel feature space to input space |
| $d$ | Number of variables (nodes) in the causal graph |
| $K$ | Number of EM iterations |
| $k$ | Current EM iteration index |
| $\alpha$ | Exponential smoothing coefficient for weight averaging |
| $\gamma, \beta$ | Thresholds for graph binarization and pruning significance |
| $p$ | Number of random Fourier features (RFF dimension) |
| $n$ | Number of independent multivariate time-series samples |

### A.3 Linear Causal Discovery as a Special Case

In domains where linear assumptions are valid or preferred for interpretability, ReTimeCausal can recover causal graphs using sparse linear regression. Each variable is regressed on its candidate parents:

$$\mathbf{X}^t = \sum_{\tau=1}^{L} \mathbf{W}^{(\tau)} \mathbf{X}^{t-\tau} + \boldsymbol{\epsilon}^t, \tag{12}$$

Here, $\mathbf{W}^{(\tau)} \in \mathbb{R}^{d \times d}$ denotes the weighted adjacency matrix that captures the causal influence from variables at lag $\tau$, where the entry $\mathbf{W}_{ij}^{(\tau)}$ quantifies the causal effect of variable $X_j^{t-\tau}$ on $X_i^t$. The noise vector $\boldsymbol{\epsilon}^t \in \mathbb{R}^d$ consists of mutually independent components, each assumed to be independent of the predictors.

Unlike the nonlinear case, where kernel methods are required to model complex dependencies in a high-dimensional space, linear relationships are directly represented in the original feature space. This enables efficient recovery of the causal graph using sparse linear regression techniques.

A common approach is to formulate causal discovery as a sparse optimization problem by minimizing the reconstruction error with an $\ell_1$-penalty to enforce sparsity in the estimated graph:

$$\min_{\mathbf{W}} \sum_{t=1}^{T} \|\mathbf{X}^t - \sum_{\tau=1}^{L} \mathbf{X}^{t-\tau} \mathbf{W}^{(\tau)}\|_2^2 + \lambda \sum_{\tau=1}^{L} \|\mathbf{W}^{(\tau)}\|_1. \tag{13}$$

where $\mathbf{X}^t \in \mathbb{R}^{n \times d}$ is the observed data matrix, $\mathbf{W}^{(\tau)} \in \mathbb{R}^{d \times d}$ is the weighted adjacency matrix representing causal coefficients, and $\lambda > 0$ controls the trade-off between fit and sparsity.

This formulation is equivalent to solving d separate Lasso regression problems, one for each variable conditioned on the others, and is computationally efficient in high-dimensional settings. Moreover, under standard assumptions (e.g., causal sufficiency, acyclicity, and additive noise), such methods can consistently identify the correct causal skeleton.

Nevertheless, when the true system exhibits nonlinear or interaction effects, purely linear models may yield biased or incomplete causal interpretations. Thus, linear models are best suited either as a baseline or in domains where linearity is a reasonable approximation.

### A.4 Graph Pruning

The thresholding step in Section 4.2 produces an initial binary causal graph $\mathcal{G}_S$ based on regression weights. However, in high-dimensional or noisy settings, this graph may still contain spurious or unstable edges—particularly when the sample size is limited or the causal signal is weak. To address this, we introduce a model-specific pruning module tailored to the functional nature of the underlying data—linear or nonlinear. This post-processing step employs a unified pruning threshold $\beta$ to filter out statistically insignificant or weakly contributing edges, thereby refining the causal graph and improving its interpretability, stability, and generalization.

**Nonlinear setting.** For nonlinear data modeled via kernel regression, we adopt a pruning approach based on Causal Additive Models (CAM) (Bühlmann et al., 2013). This method fits generalized additive models (GAMs) to each target variable and tests the significance of each input variable (including lagged parents), removing those with $p$-values exceeding a predefined threshold $\beta$. Although CAM is originally designed for static data, we apply it independently to each time-lagged parent set, leveraging its ability to capture smooth nonlinear influences while enforcing structural sparsity.

**Linear setting.** For linearly modeled systems, pruning is performed by applying a coefficient-based threshold to the absolute values of the estimated regression weights. Specifically, an edge $X_j^{t-\tau} \rightarrow X_i^t$ is retained only if $|W_{i,j}^{(\tau)}| > \beta$. This simple yet effective criterion eliminates weak dependencies that are likely to result from noise or overfitting, and is consistent with sparsity-inducing strategies used in prior DAG learning work (Zheng et al., 2018).

These pruning mechanisms act as the final refinement step in the ReTimeCausal pipeline, suppressing false positives and clarifying structural dependencies. The resulting pruned graph $\mathcal{G}_S$ is more parsimonious, statistically grounded, and better suited for interpretation and downstream applications. By separating pruning from initial graph construction, we enable modular design and allow for the incorporation of task- or domain-specific pruning strategies.

## A.5 MOTIVATION AND ALGORITHMIC FRAMEWORK

The design of ReTimeCausal is motivated by the limitations of impute-then-discover pipelines under irregular and incomplete time series. Traditional methods either assume complete observations or perform imputation independently of the causal structure, which can introduce spurious dependencies or distort temporal dynamics. To address this, ReTimeCausal performs causal discovery and missing data imputation in a joint, iterative manner.

Specifically, ReTimeCausal integrates the Additive Noise Model (ANM) into an EM-style optimization framework. At each iteration, it imputes missing values based on current causal estimates (E-step) and updates the structure through sparse regression and pruning (M-step). This integration ensures that imputation respects the evolving causal graph, and that structure learning adapts to data refinement over time.

The algorithm supports both linear and nonlinear causal mechanisms, accommodates arbitrary sampling patterns, and yields interpretable, lag-specific graphs. To improve convergence robustness, we introduce exponential weight smoothing and apply post-hoc pruning based on causal independence principles. Algorithm 1 presents the pseudocode of ReTimeCausal.

---

**Algorithm 1:** ReTimeCausal: Learning Causal Structures from Irregular Time Series

---

**Input:** Incomplete time series $\mathcal{D} = \{\mathbf{X}_o, \mathbf{X}_m\}$ with missing values; Initial regression weights $\mathbf{W}$ and neural estimators $\{f_i^{(0)}\}_{i=1}^d$

**Output:** Estimated causal graph $\mathcal{G}_S^{(K)}$ and functions $\{f_i^{(K)}\}$

1 **for** $k \leftarrow 1$ **to** $K$ **do**

    /* E-step:  Impute missing values using current estimators $\{f_i^{(k-1)}\}$ */

2     $\hat{\mathbf{X}}^{(k)} \leftarrow \text{Impute}(\mathbf{X}_o, \{f_i^{(k-1)}\})$ ;

    /* M-step:  Train neural estimators $\{f_i^{(k)}\}$ on imputed data $\hat{\mathbf{X}}^{(k)}$ */

3     **for** $i \leftarrow 1$ **to** $d$ **do**

4         Compute loss: $\mathcal{L}_i = \sum_t \left\| X_i^t - \sum_\tau \mathbf{W}^{(k,\tau)} f_i^{(k)}(\mathbf{Pa}_i^t) \right\|^2 + \lambda \|\mathbf{W}\|_1$ ;

5         Update $f_i^{(k)}$;

6     **end**

    /* Weight Smoothing */

7     $\mathbf{W}^{(k)} \leftarrow \alpha \mathbf{W}^{(k-1)} + (1-\alpha)\hat{\mathbf{W}}^{(k)}$ ;

    /* Graph Extraction */

8     $\hat{\mathcal{G}}_S^{(k)} \leftarrow \text{Binarization}(\mathbf{W}^{(k)}, \gamma)$ ;

    /* Prune Spurious Edges */

9     $\mathcal{G}_S^{(k)} \leftarrow \text{CAM}(\mathcal{D}, \hat{\mathcal{G}}_S^{(k)}, \beta)$ or $\text{LinReg}(\mathcal{D}, \hat{\mathcal{G}}_S^{(k)}, \beta)$ ;

10 **end**

11 **return** $\mathcal{G}_S^{(K)}, \{f_i^{(K)}\}$

---

Under mild conditions (bounded noise, identifiable mechanisms, compact domain), the learned functions $f_i$ converge in $\ell_1$ norm, and the estimated graph $\mathcal{G}_S^{(K)}$ converges in probability to the true summary graph $\mathcal{G}_S^*$.

### A.6 Further Discussion on the Assumptions Used in This Paper

In this section, we provide further discussion on the assumptions made in our framework and clarify their roles in both modeling and theoretical analysis. While these assumptions are standard in the literature on time series causal discovery, understanding their necessity and scope of applicability is important for both the theoretical soundness and practical flexibility of our method.

#### A.6.1 Finite-Order Markov Property and Temporal Modeling

Assumption 3.3 (Finite-Order Markov Property) states that each variable at time t depends only on a finite number of past variables. This assumption is fundamental for representing temporal dependencies through sparse regression over a fixed lag window, which underpins the structure learning component of our method. Without such an assumption, modeling would require potentially infinite memory, making both estimation and interpretation infeasible. In practice, many real-world time series systems—such as financial markets, neural recordings, or climate systems—exhibit limited temporal dependency horizons, often well approximated by first- or second-order Markov models. While our method is formulated using a fixed finite order, it is flexible enough to handle higher-order dependencies by increasing the lag window, which is straightforward to implement and evaluate in practice.

#### A.6.2 Temporal Consistency and the Validity of Summary Graphs

Assumption 3.4 (Consistency Throughout Time) requires that the causal relationships between variables do not change across time. This assumption is essential for defining a well-posed summary causal graph: if causal relationships were to drift over time, then aggregating them into a single static structure would be ill-defined or misleading. In many applications, such as analyzing causal relations in physiological systems (e.g., EEG data) or economic indicators, this assumption holds reasonably well over moderate time spans. However, we acknowledge that real-world systems may exhibit nonstationarities. In such cases, while our method can still be applied heuristically (e.g., in sliding windows), the theoretical guarantees derived under this assumption may no longer hold, and further investigation into time-varying extensions would be necessary.

#### A.6.3 Causal Sufficiency and the Impact of Latent Confounders

Assumption 3.5 (Causal Sufficiency) postulates that all common causes of observed variables are themselves observed. This is a standard assumption in many causal discovery frameworks, as it ensures that the observed statistical dependencies are not confounded by hidden variables. In our work, this assumption is particularly important for the consistency result in Proposition 1, which relies on the observed data faithfully representing the full causal structure. In practice, however, the presence of latent confounders is often unavoidable, especially in domains such as biology or social sciences. While our method does not explicitly account for hidden variables, it can be extended with techniques from the literature on causal discovery with latent confounding (e.g., FCI-based methods), which we leave as future work.

#### A.6.4 Faithfulness Assumption in Sparse Regression-Based Causal Discovery

Assumption 3.6 (Faithfulness) ensures that the statistical dependencies observed in the data correspond exactly to the causal structure, meaning that no causal relationships are masked by precise cancellation of effects. This assumption is necessary for identifying the true causal graph using statistical methods such as sparse regression, which relies on inferring structure from conditional independence patterns. While faithfulness is a strong condition, it is widely adopted in causal inference, and violations typically occur only under specific and rare parameterizations. In applied scenarios, approximate faithfulness is often sufficient to yield useful and interpretable causal graphs, especially when combined with regularization strategies that promote robustness to sampling noise.

#### A.6.5 Non-Gaussianity in Additive Noise Models and Identifiability

Finally, regarding the non-Gaussianity of noise, we emphasize that this is not a global requirement of our method. It is only invoked in the identifiability analysis of nonlinear additive noise models (ANMs), where classical results show that causal direction can be determined when the noise is

non-Gaussian and independent of the cause. This assumption enables stronger theoretical guarantees in Section 4.2, particularly when learning causal direction from observational data. However, in practical applications, our method remains applicable even in Gaussian noise settings; in such cases, identifiability may be weaker, but the structure learning and estimation methods still produce meaningful and informative causal summaries. In real-world data—such as sensor networks, economic systems, or biological processes—non-Gaussian noise is often present due to heterogeneity, measurement error, or nonlinear dynamics, making this assumption not only theoretically useful but also practically reasonable in many settings.

## A.7 CONVERGENCE ANALYSIS

The core of ReTimeCausal lies in its iterative EM style optimization, where missing data is imputed based on current causal estimates, and structural functions are subsequently updated via sparse regression. Understanding the convergence behavior of this process is essential to ensure that the learned causal graph stabilizes and aligns with the underlying true structure as data grows.

This iterative learning process can be understood as a two-level optimization scheme: (i) an outer loop that alternates between imputing missing data and updating the causal graph (EM iterations), and (ii) an inner loop that solves kernel-based sparse regression problems to refine the structural functions. We analyze the convergence of both components under standard assumptions.

At each iteration, the imputed data $\hat{\mathbf{X}}_m^{(k)}$ is mapped into a kernel feature space, enabling the estimation of complex nonlinear structural functions via convex optimization. This forms a two-level learning process: (i)The outer loop (EM) alternates between imputing data and updating structure, and (ii)the inner loop (kernel regression) refines structural functions in a kernel-embedded space.

Under standard assumptions—such as bounded kernels, compact input domains, and noise terms that are independent of the parent variables—the learned structural functions converge to the true $f_i$ in the $L^2$ norm. Specifically:

$$\forall \epsilon > 0, \ \exists \ f_i^{(\text{kernel})} \in \mathcal{H}_\kappa \quad \text{s.t.} \quad \|f_i - f_i^{(\text{kernel})}\|_{L^2} < \epsilon$$

Furthermore, sparse kernel regression ensures that the estimated graph $\hat{\mathcal{G}}_S$ converges in probability to the true summary graph $\mathcal{G}_S^*$ as $n \to \infty$, assuming faithfulness (Rosasco et al., 2004; Spirtes et al., 2001).

### A.7.1 PROOF OF PROPOSITION 1

**Proposition 1 (Structural Consistency)** *Let the variable sequence $\{\mathbf{X}^1, ..., \mathbf{X}^t\}$ be generated by a finite-order Markov process that satisfies Assumptions 3.2–3.6. If the regularization and thresholding parameters used in the algorithm decrease appropriately with the sample size, and the smoothing weight remains within a fixed range, then the estimated summary causal graph $\hat{\mathcal{G}}_S^{(k)}$ converges structurally to the true graph $\mathcal{G}_S^*$, satisfying:* $\lim_{k\to\infty} \lim_{n\to\infty} \Pr\left(\hat{\mathcal{G}}_S^{(k)} = \mathcal{G}_S^*\right) = 1.$

To establish Proposition 1, we outline the proof of structural consistency by decomposing the argument into four key components. We assume throughout that Assumptions 3.2 (Ignorable Missingness), 3.3 (finite-order Markovity), 3.4 (Consistency Throughout Time), 3.5 (causal sufficiency), and 3.6(faithfulness) hold.

**Step 1: Consistency of EM under Ignorable Missingness.** Under the assumption 3.2 and model identifiability, the classical Expectation-Maximization algorithm yields asymptotically consistent maximum likelihood estimates (MLEs) of the model parameters (Dempster et al., 1977; Wu, 1983). That is, as the sample size $n \to \infty$ and iteration number $k \to \infty$, we have

$$\hat{\theta}^{(k)} \xrightarrow{p} \theta^* \quad \text{and} \quad \hat{\mathbf{X}}^{(k)} \xrightarrow{p} \mathbf{X}^*,$$

where $\theta^*$ is the true parameter and $X^*$ denotes the completed data under the true model.

**Step 2: Consistency of Nonlinear Estimators.** For each structural equation $X_i^t = f_i(\mathbf{Pa}_i^t) + \epsilon_i^t$, assume $f_i$ belongs to a function class $\mathcal{F}$ with universal approximation capacity, such as Reproducing

Kernel Hilbert Spaces (RKHS) or sufficiently wide neural networks. Then, by standard learning theory (e.g., (Schölkopf & Smola, 2002; Györfi et al., 2002)), for any $\varepsilon > 0$, there exists an estimator $f_i^{(k)} \in \mathcal{F}$ such that

$$\|f_i^{(k)} - f_i^*\|_{L^2} < \varepsilon \quad \text{as } n \to \infty.$$

Hence the learned structural functions converge to the ground truth, provided the imputations are accurate (Step 1).

**Step 3: Consistency of Sparse Structure Recovery.** Let $\mathcal{G}_S^*$ be the true causal graph and $\hat{\mathcal{G}}_S^{(k)}$ the estimate at iteration $k$. Assume (i) the structural functions admit identifiable ANMs, (ii) regularity of the design (e.g., the irrepresentability condition (Wainwright, 2009)), and (iii) appropriate regularization (e.g., $\lambda_n \to 0$ with $\sqrt{n}\lambda_n \to \infty$). Then sparse regression on $\hat{\mathbf{X}}^{(k)}$ achieves support recovery:

$$\Pr\left(\text{supp}(\hat{\mathbf{W}}^{(k)}) = \text{supp}(\mathbf{W}^*)\right) \to 1 \quad \text{as } n \to \infty.$$

Under Assumption 3.6 (faithfulness), the recovered supports correspond to the true causal edges.

**Step 4: Graph Stabilization via Smoothed Estimation and Thresholded Recovery.** Building on the support recovery guarantee from Step 3, we now analyze the impact of exponential smoothing and entrywise thresholding on graph stability. In addition to support consistency, standard results from high-dimensional sparse estimation theory (Wainwright, 2009) imply that, under the same regularity assumptions, the estimation error in $\ell_1$ norm satisfies

$$\|\hat{\mathbf{W}}^{(k)} - \mathbf{W}^*\|_1 = o(\gamma_n) \quad \text{as } n \to \infty,$$

where $\gamma_n$ is the thresholding parameter. This result provides a foundation for analyzing the behavior of the smoothed weights $\mathbf{W}^{(k)}$, which are computed via

$$\mathbf{W}^{(k)} = \alpha \mathbf{W}^{(k-1)} + (1-\alpha) \hat{\mathbf{W}}^{(k)}.$$

subtract $\mathbf{W}^* = \alpha \mathbf{W}^* + (1-\alpha)\mathbf{W}^*$ and apply the triangle inequality to obtain the recursion

$$\left\|\mathbf{W}^{(k)} - \mathbf{W}^*\right\|_1 \leq \alpha \left\|\mathbf{W}^{(k-1)} - \mathbf{W}^*\right\|_1 + (1-\alpha)\left\|\hat{\mathbf{W}}^{(k)} - \mathbf{W}^*\right\|_1.$$

Unrolling the recursion yields, for any $k \geq 1$,

$$\left\|\mathbf{W}^{(k)} - \mathbf{W}^*\right\|_1 \leq \alpha^k \left\|\mathbf{W}^{(0)} - \mathbf{W}^*\right\|_1 + (1-\alpha)\sum_{j=1}^{k} \alpha^{k-j}\left\|\hat{\mathbf{W}}^{(j)} - \mathbf{W}^*\right\|_1.$$

Since $\|\hat{\mathbf{W}}^{(j)} - \mathbf{W}^*\|_1 = o(\gamma_n)$ (uniformly over fixed $j \leq k$) and $\alpha \in [0,1)$, it follows that $\|\mathbf{W}^{(k)} - \mathbf{W}^*\|_1 = o(\gamma_n)$ as $n \to \infty$ for each fixed $k$, and in fact $\mathbf{W}^{(k)} \to \mathbf{W}^*$ geometrically in $k$ up to the vanishing estimation error.

To ensure correct support recovery under thresholding, we assume a symmetric margin condition: there exists a fixed constant $m > 0$ such that

$$\min_{(i,j)\in\text{supp}(\mathbf{W}^*)} |W_{ij}^*| > m.$$

This margin separates nonzero and zero coefficients by at least $m$, enabling thresholding at levels $\gamma_n \in (0, m)$ to recover the correct support with high probability.

With a symmetric margin $m > 0$ and a threshold $\gamma_n \in (\gamma_0, \gamma_{\min})$, this implies support stabilization for large $k$ and $n$:

$$\lim_{k\to\infty} \text{supp}(\mathbf{W}^{(k)}) = \text{supp}(\mathbf{W}^*) \quad \text{and} \quad \lim_{k\to\infty}\lim_{n\to\infty} \Pr\left(\hat{\mathcal{G}}_S^{(k)} = \mathcal{G}_S^*\right) = 1.$$

A.7.2 CONVERGENCE IN REALISTIC SETTINGS

This section analyzes the error convergence of ReTimeCausal under realistic constraints—namely, finite samples and a finite number of EM iterations. In contrast to the idealized setting considered in Appendix A.5.1, we focus on three primary sources of error that arise in practice:(i) statistical

estimation error due to limited sample size, (ii) kernel approximation error introduced by finite-dimensional feature mappings, and (iii) optimization residual caused by incomplete EM convergence.

Building on the assumptions established in the main text, we additionally introduce two mild conditions commonly used in high-dimensional EM analysis: (i) geometric absolute regularity to ensure concentration bounds for dependent observations (Merlevède et al., 2009), and (ii) gradient stability and restricted strong concavity (RSC) to guarantee local contractivity of the EM operator (Balakrishnan et al., 2014).

Under this framework, we formally characterize the evolution of the imputation error $e_k$ at each EM iteration. By leveraging the sparsity structure induced by feature-to-input projection, we derive a quantitative bound on the Structural Hamming Distance (SHD), expressed in terms of sample size, feature dimension, and the number of EM iterations.

**Theorem 1 (Convergence Guarantee under Finite Sample Size and Iterations)** *Suppose the population EM operator satisfies local contractivity with rate $\rho_{\mathrm{pop}} < 1$ under mild curvature and stability assumptions. Let $\lambda$ be a regularization level proportional to $\sqrt{\log(\widetilde{d})/n}$, and let $\tau_n$ denote the finite-sample deviation term. Then there exist constants $C_1 > 0$ such that*

$$\mathrm{SHD}\Big(\mathcal{G}^{(k)}, \mathcal{G}^*\Big) \;\leq\; \frac{C_1}{m} \left( s\sqrt{\frac{\log(\widetilde{d})}{n}} + \delta_p + \rho^k e_0 + \frac{\tau_n}{1 - \rho} \right).$$

Before proceeding with the proof, we recall the key assumptions and notation:

- Let $\hat{\mathbf{X}}^{(k)}$ denote the imputed data after the $k$-th EM iteration, and define the imputation error $e_k := \|\hat{\mathbf{X}}_m^{(k)} - \mathbf{X}_m^*\|_\infty$.
- Let $\widetilde{d} := d^2 L$ denote the total number of pairwise temporal interactions across $d$ variables with maximum lag $L$.
- The regularization level is set as:

$$\lambda \asymp \sigma\sqrt{\frac{\log(\widetilde{d})}{n}}, \quad \text{with} \quad n \gtrsim s\log(\widetilde{d}),$$

  where $s$ is the sparsity level of the true coefficient matrix $\mathbf{W}^*$ and $\sigma^2$ is the sub-Gaussian proxy variance (Bickel et al., 2008; Negahban et al., 2010).
- Feature map $\phi : \mathbb{R}^d \to \mathbb{R}^p$ is $L_\phi$-Lipschitz and uniformly approximates a bounded kernel with error $\delta_p \to 0$ as $p \to \infty$ (Rahimi & Recht, 2007; Rudi & Rosasco, 2017; Steinwart & Christmann, 2008).
- The projection from feature space to input space is given by $\Pi := \Phi^\top$, with $\|\Pi\|_{1 \to 1} \leq C_\Pi$.
- The margin condition assumes that true edges have weight at least $\gamma_{\min}$ and false edges at most $\gamma_0$, with threshold $\gamma \in (\gamma_0, \gamma_{\min})$ and

$$m := \min\{\gamma_{\min} - \gamma, \; \gamma - \gamma_0\} > 0.$$

  This condition is natural in the ground truth graph, where spurious edge weights are strictly smaller than the weakest true edge.

**Step 1: $\ell_1$ estimation bound for the projected M-step.** We first establish an upper bound on the $\ell_1$-norm error between the estimated coefficients $\hat{\mathbf{W}}^{(k)}$ and the true sparse tensor $\mathbf{W}^*$ after $k$ EM iterations. By applying standard techniques for high-dimensional regression under absolute regularity dependence (e.g., primal–dual witness arguments and RE conditions) (Bickel et al., 2008; Basu & Michailidis, 2015), and using the stability of the RE constant under bounded imputation error $e_k$, we obtain:

$$\big\|\hat{\mathbf{W}}^{(k)} - \mathbf{W}^*\big\|_1 \leq \frac{4s\lambda}{\phi_0^2} + c_0 e_k + c_0 \delta_p.$$

Here, $c_0$ is a constant depending only on smoothness and projection parameters (e.g., $L_f, L_\phi, \phi_0, C_\Pi$), and $\delta_p$ quantifies the kernel approximation error from the finite-dimensional feature map (Rahimi & Recht, 2007; Rudi & Rosasco, 2017).

**Step 2: Bounding the iteration error $e_k$ via EM recursion.** Next, we analyze the evolution of the imputation error $e_k$ over EM iterations. Using the local contractivity of the population EM operator (under curvature and gradient stability assumptions) (Balakrishnan et al., 2014), combined with a high-probability uniform deviation bound between the sample and population operators (controlled via $\tau_n$) (Merlevède et al., 2009), we obtain the recursion:

$$e_{k+1} \le \rho \, e_k + \tau_n \quad \Rightarrow \quad e_k \le \rho^k e_0 + \frac{\tau_n}{1 - \rho},$$

where $\rho \in (\rho_{\text{pop}}, 1)$ and $\tau_n = O\left(\sqrt{\frac{\log(\tilde{d})}{n}}\right)$ captures the statistical fluctuation due to absolute regularity samples.

**Step 3: Converting coefficient error to SHD via thresholding.** Finally, we connect the coefficient estimation error to the Structural Hamming Distance (SHD) via the thresholding margin condition. Suppose all true edges have weights at least $\gamma_{\min}$, and all spurious edges have magnitudes at most $\gamma_0$. Then, thresholding at level $\gamma \in (\gamma_0, \gamma_{\min})$ correctly separates true and false edges, provided that the entrywise estimation error satisfies

$$\max_{i,j} |\hat{W}_{ij}^{(k)} - W_{ij}^*| < m.$$

This condition reflects a standard minimal signal strength assumption (also referred to as a margin condition), which is commonly used to ensure support recovery consistency under thresholding (Zhang & Zhang, 2014; Bühlmann & Van De Geer, 2011). Consequently, the Structural Hamming Distance (SHD) is upper bounded by the total $\ell_1$ error divided by $m$, as each unit of SHD corresponds to at least $m$ of deviation in the coefficient matrix:

$$\text{SHD}\left(\hat{\mathcal{G}}^{(k)}, \mathcal{G}^*\right) \le \frac{1}{m} \left\| \hat{\mathbf{W}}^{(k)} - \mathbf{W}^* \right\|_1.$$

**Step 4: Putting it all together.** Combining the results from Steps 1–3 and substituting the explicit form of $\lambda$ and $\tau_n$, we obtain the desired upper bound on the structural Hamming distance:

$$\text{SHD}\left(\hat{\mathcal{G}}^{(k)}, \mathcal{G}^*\right) \le \frac{C_1}{m} \left( s\sqrt{\frac{\log(\tilde{d})}{n}} + \delta_p + \rho^k e_0 + \frac{\tau_n}{1 - \rho} \right),$$

which corresponds to the form presented in Theorem 1.

### A.7.3 PRACTICAL LIMITATIONS AND CONVERGENCE BEHAVIOR

While the theoretical analysis establishes the asymptotic consistency of ReTimeCausal under standard assumptions, it is important to examine how these guarantees translate into practical settings with finite data, model approximations, and optimization heuristics. This section discusses several implementation-level factors that may influence convergence behavior in real-world applications.

First, although the EM-style method provides a principled framework for joint imputation and structure learning, its convergence rate can vary significantly depending on data characteristics such as the missingness rate ($r$), the degree of nonlinearity in causal functions $f_i(\cdot)$, and temporal sparsity. In our formulation, the E-step computes the imputed values based on current estimators.

In practice, we match this distribution to the residuals observed during function fitting (e.g., Laplace or empirical noise), ensuring compatibility with a wide range of noise models. High missingness rates ($r > 0.6$) or heavy-tailed noise can slow down EM convergence and may require additional iterations for structure stabilization.

Second, in the M-step, each causal function $f_i$ is estimated using sparse regression in a kernel-induced space. To promote both convergence and interpretability, we apply $\ell_1$-penalized regression over lagged features, as previously introduced in Eq. 7. This formulation encourages sparsity across temporal lags while enabling the recovery of nonlinear dynamics via the kernel mapping $\phi(\cdot)$. However, since the objective becomes non-convex when neural or RBF kernels are used, convergence to a global minimum is not guaranteed. We mitigate this through two stabilizing techniques:

- **Exponential smoothing**: We maintain a smoothed estimate of regression weights across iterations: $\mathbf{W}^{(k)} = \alpha \mathbf{W}^{(k-1)} + (1-\alpha)\hat{\mathbf{W}}^{(k)}$;

- **Threshold-based pruning**: Weak edges are removed using a lag-specific cutoff $\gamma > 0$, reducing variance in graph estimates and improving interpretability.

Third, although the method relies on alternating updates, it does not involve adversarial training or black-box inference. This enhances empirical robustness and enables direct control over convergence via interpretable hyperparameters such as $\lambda$, $\alpha$, and $\gamma$.

Although the convergence speed and quality of ReTimeCausal may be affected by factors such as initialization and model complexity, the framework exhibits stable iterative behavior across a wide range of settings. Its modular architecture facilitates integration with more efficient optimization strategies or domain-specific priors, enhancing its adaptability to diverse real-world applications.

### A.8 DISCUSSION OF COMPARED BASELINES

To evaluate the performance of ReTimeCausal, we compare it against several widely used causal discovery algorithms that represent mainstream methodological paradigms. Specifically, we consider the following baselines:

**PCMCI** (Runge et al., 2019b) is a constraint-based method tailored for high-dimensional time series. It combines conditional independence testing with a temporal extension of the PC algorithm, providing statistical control under assumptions of causal sufficiency and time-lagged dependence.

**DYNOTEARS** (Pamfil et al., 2020) is a score-based approach that extends NOTEARS to dynamic settings. It optimizes a smooth objective with acyclicity constraints and sparse regularization, enabling the estimation of time-varying causal graphs. Since the original DYNOTEARS is designed for linear dynamics, we include a kernelized variant denoted as DYNOTEARS$^*$ for comparison in nonlinear settings. DYNOTEARS$^*$ replaces the linear regression module with our kernelized sparse regression method, allowing it to model nonlinear dependencies and serve as a more appropriate baseline for evaluating ReTimeCausal in nonlinear regimes.

**CUTS+** (Cheng et al., 2024a) is a recent iterative framework designed for irregular time series. It leverages uncertainty-aware structure learning, message-passing neural networks, and cycle-consistent data augmentation to enhance robustness and scalability.

**Rhino** (Gong et al., 2023) is a deep causal discovery method for time series. It models nonlinear relations, instantaneous effects, lagged effects, and history-dependent noise. It gives a flexible way to learn temporal causal graphs under complex real-world dynamics. Rhino serves as a strong baseline when the data have nonlinear or fast-changing patterns.

**AERCA** (Han et al., 2025) is a root-cause analysis method that also contains a built-in Granger causal discovery module. It models exogenous variables and their normal patterns. It then uses deviations in these variables to identify possible root causes. Because AERCA learns a Granger-style causal structure as part of its process, it provides an additional reference point for evaluating causal discovery in our setting.

Most of these methods do not handle missing data and require fully observed inputs. To ensure fair comparison, we apply **TimeMixer** (Wang et al., 2024) as a preprocessing step. TimeMixer decomposes a time series into multiple resolutions and uses learnable mixing operations. It captures short- and long-term patterns and gives high-quality imputations for downstream causal discovery."

**Evaluation Metrics.** To quantitatively compare performance, we report three standard metrics commonly used in causal discovery:

- **SHD (Structural Hamming Distance)**: Measures the number of edge additions, deletions, or reversals needed to transform the estimated causal graph into the ground truth. Lower values indicate better structural accuracy.

- **SID (Structural Intervention Distance)**: Captures the number of incorrect causal inferences under intervention. It reflects the mismatch between predicted and true interventional distributions, with lower values indicating better causal robustness.

- **F1 Score**: The F1 score represents the harmonic mean of precision and recall, evaluated on the predicted causal edges. It quantifies the balance between sensitivity and specificity, with higher values indicating more accurate and reliable structure recovery.

## A.9 ADDITIONAL EXPERIMENTS AND ABLATION STUDIES

To further evaluate the robustness, effectiveness, and design choices of the ReTimeCausal framework, we conduct a series of additional experiments and ablation studies. While the main results in Section 5 demonstrate strong overall performance across various settings, it remains important to understand how individual components of the model contribute to its success, how it behaves under different nonlinear mechanisms, and how it scales with increasing complexity. This section presents targeted analyses that isolate specific factors such as noise injection, nonlinearity types, temporal lag orders, and real-world generalization, thereby providing a deeper empirical characterization of the framework's behavior.

### A.9.1 EFFECT OF NOISE INJECTION

To assess the role of noise injection in ReTimeCausal, we compare the full model with a variant that disables this component. As shown in Table 5, removing noise injection results in a dramatic drop in F1 score (from 0.976 to 0.159) and a significant increase in SHD (from 1 to 212). These results highlight that noise injection is essential for accurate structure recovery.

Table 5: Robustness Comparison under Functionally Distinct Nonlinear Settings

| Methods | SIN-laplace-20-20-1 | | | TANH-laplace-20-20-1 | | |
|---|---|---|---|---|---|---|
| | SHD | SID | F1 | SHD | SID | F1 |
| PCMCI+TimeMixer | 55 | 12 | 0.321 | 25 | 34 | 0.194 |
| PCMCI+EM | 22 | 35 | 0.154 | 21 | 33 | 0.276 |
| DYNOTEARS*+TimeMixer | 78 | 21 | 0.204 | 36 | 40 | 0.100 |
| Rhino+TimeMixer | 51 | 29 | 0.164 | 77 | 29 | 0.115 |
| AERCA+TimeMixer | 54 | 34 | 0.100 | 176 | 15 | 0.120 |
| CUTS+ | 197 | 11 | 0.116 | 66 | 25 | 0.154 |
| ReTimeCausal w/o Noise | 212 | **0** | 0.159 | 185 | **0** | 0.178 |
| ReTimeCausal | **1** | **0** | **0.976** | **1** | 1 | **0.974** |

From a theoretical perspective, noise injection serves as a form of regularization in both optimization and statistical estimation. In the context of Expectation-Maximization, missing data can cause the algorithm to converge prematurely to suboptimal local minima, particularly when early-stage imputations are biased or overconfident. By perturbing the imputed residuals with small, zero-mean noise, we prevent the regression step from overfitting to inaccurate imputations, thereby promoting exploration and reducing the risk of early convergence to poor structures.

In kernel regression, this mechanism additionally mitigates overfitting in high-dimensional feature spaces, where exact interpolation of sparse or noisy points can lead to unstable estimates. Injecting noise smooths the learned functions in the RKHS, improving generalization.

Moreover, from a statistical testing viewpoint, our pruning method relies on testing independence between the regression residuals and potential parents. However, if residuals are deterministically derived from imperfect imputations, the independence assumption may be violated. Noise injection restores this independence in expectation, making the downstream conditional independence tests more statistically valid.

### A.9.2 ADAPTABILITY TO NONLINEAR COMPLEXITY

To assess the adaptability of ReTimeCausal to increasing functional nonlinearity, we evaluate performance across synthetic datasets generated using distinct nonlinear structural functions, including $\sum \mathbf{Sin}(\mathbf{Pa}_i^t)$ and $\sum \mathbf{Tanh}(\mathbf{Pa}_i^t)$. All datasets share the same underlying causal graph and noise distribution. As shown in Table 5, ReTimeCausal maintains high accuracy across a wide spectrum of nonlinearities, demonstrating its expressive capacity and generalization.

Table 6: Causal Graph Recovery Performance Under Varying Maximum Lag Orders

| Methods | L = 2 | | | L = 3 | | | L = 4 | | |
|---|---|---|---|---|---|---|---|---|---|
| | SHD | SID | F1 | SHD | SID | F1 | SHD | SID | F1 |
| PCMCI+TimeMixer | 19 | 6.0 | 0.230 | 25 | 23.0 | 0.144 | 32 | 16.0 | 0.130 |
| PCMCI+EM | 39 | 11.0 | 0.316 | 17 | 23.0 | 0.414 | 27 | 33.0 | 0.069 |
| DYNOTEARS+TimeMixer | 9 | 13.3 | 0.640 | 8 | 15.0 | 0.765 | 21 | 6.3 | 0.439 |
| Rhino+TimeMixer | 40 | 12.0 | 0.200 | 71 | **3.0** | 0.202 | 6.0 | 79 | 0.190 |
| AERCA+TimeMixer | 42 | 13.0 | 0.160 | 50 | 18.0 | 0.167 | 44 | 28.0 | 0.120 |
| CUTS+ | 72 | 3.0 | 0.200 | 233 | 6.0 | 0.127 | 252 | **5.0** | 0.137 |
| ReTimeCausal | **2** | **0.3** | **0.941** | 5 | 10.3 | **0.865** | 17 | 6.3 | **0.730** |

### A.9.3 MULTI-ORDER LAG ANALYSIS

To evaluate the ability of ReTimeCausal to recover multi-order lagged causal structures, we conduct a targeted experiment on a synthetic linear time series dataset with 10 variables and 10 edges. The dataset includes 60% randomly missing values, simulating a challenging irregular setting.

We generate and test three variants of the dataset with different maximum lag orders: $L = 2$, $L = 3$, and $L = 4$. For each configuration, we run ReTimeCausal and evaluate the structure recovery performance using standard metrics such as SHD, SID, and F1 score. The results are summarized in Table 6. Additionally, Figure 3 visualizes the recovered second-order lagged causal graph, demonstrating that our regression-based structure learning accurately captures the underlying dependencies, while the pruning strategy effectively removes spurious edges, resulting in a clean and interpretable causal structure. All reported metrics are computed on the temporally unrolled graphs that merge all lag orders, which may lead to relatively large SHD values (e.g., exceeding 100) due to the expanded graph size. Since CUTS+ only outputs a single summary-level causal graph, we treat this graph as being shared across all lag orders during evaluation.

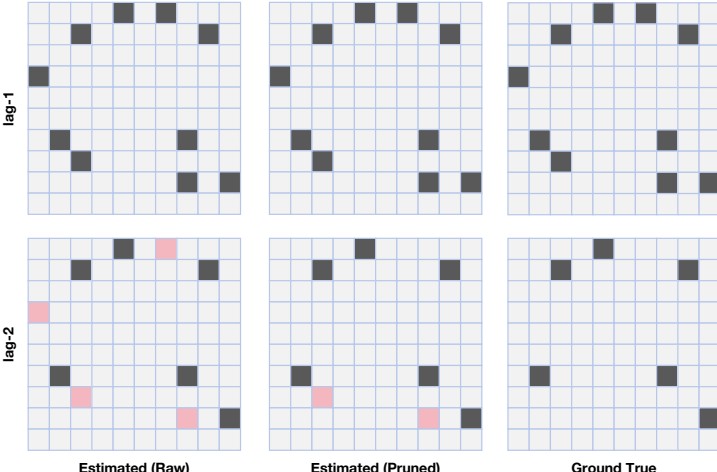

Figure 3: Visualization of recovered lagged causal graphs for the $L = 2$ setting. The top and bottom rows correspond to lag-1 and lag-2 structures, respectively. From left to right: estimated graph before pruning, pruned graph, and ground truth. Black squares denote correctly identified causal edges; red squares highlight spurious edges.

### A.9.4 EVALUATING STABILITY WITH RESPECT TO $R^2$-SORTABILITY

Recently, $R^2$-sortability has gained increasing attention as an artifact that can inadvertently boost the apparent performance of causal discovery methods (Reisach et al., 2023; Herman et al., 2025), thereby raising further concerns in time series causal discovery (Lohse & Wahl, 2025). The core idea is to use the coefficient of determination ($R^2$)—the proportion of variance in a variable ex-

plained by its direct causal parents—as a criterion for ordering variables along the causal direction. In systems where downstream variables tend to exhibit higher $R^2$ values (i.e., they can be better explained by their parents), the system is said to possess high $R^2$-sortability. Formally, $R^2$-sortability is defined as the average proportion of directed edges whose orientation is consistent with the ordering induced by $R^2$. Unlike varsortability, which is based on marginal variances, $R^2$-sortability emphasizes explanatory power, which provides a complementary perspective for characterizing data properties relevant to causal discovery.

In this section, we investigate how varying levels of $R^2$-sortability influence the stability and effectiveness of our method, and we demonstrate that its performance remains robust regardless of the degree of $R^2$-sortability.

We follow the synthetic data design principles proposed in the UUMC framework (Herman et al., 2025): (i) Unitless, (ii) Unrestricted, and (iii) Markov-Consistent. Based on the LR-gaussian-10-10-2 generation setting, we further adjust the distribution of noise standard deviations and the method for generating noise sortability to obtain a collection of synthetic datasets whose $R^2$-sortability values lie within the range $[0.2, 0.8]$. The detailed settings and corresponding sortability values of the generated datasets are summarized as follows:

Table 7: Sortability under different noise settings in UUMC synthetic data

| Level | Noise Std Range | Sorted Noise | Var-Sortability | $R^2$-Sortability |
|-------|-----------------|--------------|-----------------|-------------------|
| Low   | [0.5, 2.0]      | ×            | 0.503           | 0.272             |
| Mid   | [1.0, 1.0]      | ×            | 0.003           | 0.503             |
| High  | [0.5, 2.0]      | ✓            | 0.012           | 0.794             |

Table 7 illustrates how varying noise levels and noise sorting strategies affect both var-sortability and $R^2$-sortability. To further investigate the impact of $R^2$-sortability on the performance of causal discovery methods, we evaluate standard metrics under different levels of $R^2$-sortability. The corresponding results are reported in Table 8.

Table 8: Evaluation metrics under different $R^2$-sortability levels

| Level | SHD | SID | F1 |
|-------|-----|-----|------|
| Low   | 1   | 0   | 0.973 |
| Mid   | 0   | 0   | 1.000 |
| High  | 0   | 0   | 1.000 |

The evaluation results indicate that our method maintains robust performance across all three levels of sortability. While the performance under the low sortability condition is marginally reduced compared to the mid and high levels, it still achieves a high F1 score of 0.973. These findings suggest that the proposed approach exhibits strong resilience to variations in sortability and that its effectiveness is not contingent upon the presence of favorable sortability structures.

### A.9.5 Experiments on Additional Real-World Benchmarks

In addition to the CausalRivers evaluation, we further assess our method on two widely used benchmarks: **NetSim** and the recently released **CausalTime**. NetSim is a standard dataset for functional connectivity analysis in fMRI, providing biologically plausible synthetic time series with ground-truth networks, while accounting for hemodynamic variability and confounding effects. Following prior work, we introduce missing values through random masking with rates of 0.2 and 0.6 to emulate incomplete observations.

As shown in Table 9, ReTimeCausal achieves the best performance under both settings (F1 = 0.742 and 0.689), outperforming all competing methods. CUTS is competitive in the low-missing case but degrades substantially under higher missingness, whereas AERCA, PCMCI-based, and DYNOTEARS-based baselines deteriorate more markedly. Rhino+TimeMixer shows lowest F1

scores because Rhino depends on stable temporal patterns, while TimeMixer's coarse multiscale reconstruction introduces distortions that weaken Rhino's variational estimation.

Table 9: Performance comparison on the NetSim benchmark under different missing rates

| Methods | Sim3 with 10 nodes 15 edges | |
|---|---|---|
| | MR=0.2 | MR=0.6 |
| PCMCI+TimeMixer | 0.655 | 0.600 |
| PCMCI+EM | 0.405 | 0.408 |
| DYNOTEARS*+TimeMixer | 0.565 | 0.531 |
| Rhino+TimeMixer | 0.258 | 0.246 |
| AERCA+TimeMixer | 0.513 | 0.436 |
| CUTS+ | 0.667 | 0.571 |
| ReTimeCausal | **0.742** | **0.689** |

Beyond NetSim, we extend our evaluation to the **CausalTime** benchmark (Cheng et al., 2024b), which contains multivariate time series from ecological, financial, and medical domains. Compared to CausalRivers, CausalTime provides more diverse and realistic real-world conditions. In particular, we report results on datasets from the Medical and PM25 domain to examine performance in clinically relevant settings. As summarized in Table 10, ReTimeCausal again achieves the highest F1 scores (0.658 and 0.648 under MR = 0.2 and 0.6), demonstrating consistent robustness under high levels of missingness.

Table 10: Performance comparison on the CausalTime benchmark under different missing rates

| Methods | Medical | | PM25 | |
|---|---|---|---|---|
| | MR=0.2 | MR=0.6 | MR=0.2 | MR=0.6 |
| PCMCI+TimeMixer | 0.491 | 0.479 | 0.433 | 0.407 |
| PCMCI+EM | 0.160 | 0.207 | 0.508 | 0.264 |
| DYNOTEARS*+TimeMixer | 0.541 | 0.539 | 0.549 | 0.557 |
| Rhino+TimeMixer | 0.472 | 0.405 | 0.431 | 0.430 |
| AERCA+TimeMixer | 0.386 | 0.378 | 0.245 | 0.238 |
| CUTS+ | 0.667 | 0.571 | 0.468 | 0.444 |
| ReTimeCausal | **0.658** | **0.648** | **0.633** | **0.649** |

### A.9.6 EFFECT OF IMPUTATION–DISCOVERY INTERACTION

Figure 4 illustrates the evolution of imputation quality and causal discovery accuracy across EM iterations in Linear setting (LR-gaussian-20-20-1). The two curves exhibit a clear mutual reinforcement pattern, which constitutes the core mechanism behind our joint EM updates.

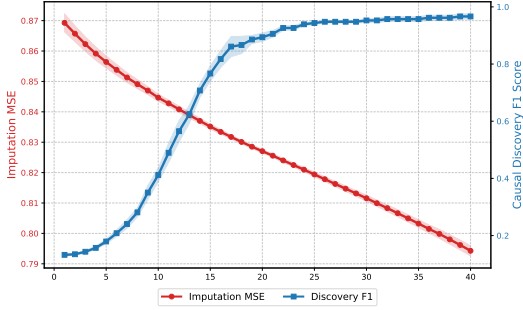

Figure 4: Mutual promotion between imputation and causal discovery during joint training. Results are averaged over 5 random seeds with shaded areas indicating standard error.

The figure shows that the imputation MSE continues to decrease throughout the iterations, while the F1 score becomes stable after roughly 20 iterations. This behavior is reasonable because reducing the imputation error requires continuously improving the predicted values, while the estimated causal structure changes only when the main effect patterns shift. Once the structure stabilizes, later iterations mainly improve prediction accuracy, which reduces the MSE but no longer alters the recovered graph.

### A.9.7 HYPERPARAMETER SENSITIVITY ANALYSIS

We assess ReTimeCausal's resilience to its important hyperparameters, such as the smoothing coefficient $\alpha$, pruning threshold $\beta$, and binarization threshold $\gamma$, in addition to the ablation studies mentioned above. We report performance on a representative nonlinear configuration (TANH–laplace–20–40–1) under 60% missingness, varying each hyperparameter's value over a wide range while holding all others constant.

Table 11: Hyperparameter Sensitivity Analysis

| Level | SHD | F1 |
|---|---|---|
| $\alpha = 0.70$ | 4 | 0.952 |
| $\alpha = 0.80$ | 5 | 0.938 |
| $\alpha = 0.90$ | 3 | 0.964 |
| $\beta = 0.01$ | 4 | 0.951 |
| $\beta = 0.05$ | 5 | 0.938 |
| $\beta = 0.10$ | 7 | 0.920 |
| $\gamma = 0.05$ | 5 | 0.938 |
| $\gamma = 0.15$ | 9 | 0.891 |
| $\gamma = 0.25$ | 9 | 0.883 |

ReTimeCausal is stable over a broad range of hyperparameter settings, as demonstrated by the results in Table 11. For values between 0.7 and 0.9, the smoothing coefficient $\alpha$ yields almost identical performance, suggesting that the exponential averaging mechanism is not sensitive to its precise selection. For the CAM pruning threshold $\beta$, a stricter setting such as $\beta = 0.01$ provides the most reliable results; because CAM removes an edge only when its p-value exceeds $\beta$, smaller values enforce stronger significance requirements and lead to more effective pruning. In contrast, larger thresholds like $\beta = 0.10$ tend to retain statistically insignificant edges and introduce false positives. The binarization threshold $\gamma$ also benefits from smaller values: thresholds around 0.05–0.15 preserve weak but genuine causal coefficients that CAM can further assess, whereas higher thresholds eliminate informative edges prematurely.

### A.10 COMPLEXITY ANALYSIS

We provide a consolidated analysis of the per-iteration computational complexity of the EM procedure. Let the batch size be $B$, sequence length $T$, number of variables $d$, lag order $L$, RFF feature dimension $p$, optimizer iterations $I$, and missing-rate $\rho$.

During the E-step, each missing entry, retrieving its lagged parent values costs $O(Ld)$, and evaluating the structural function $f_i(\cdot)$ costs $O(C_f)$. Since each $f_i$ is a 3-layer MLP with constant hidden width, we have $C_f \approx O(Ld)$. With $M = \rho BTd$ missing entries, the total E-step cost becomes $O(M(Ld + C_f)) = O(\rho BTLd^2) = O(BTLd^2)$.

During the M-step, updating all structural functions over the $V = (1 - \rho)BTd$ observed samples requires $O(V(Ld + C_f)) = O(BTLd^2)$. Kernelized sparse regression, which includes constructing random Fourier features and performing sparse optimization, incurs a cost of $O(BTLdp) + O(ILBTdp) = O(ILBTdp)$. Nonlinear CAM pruning constructs, for each variable, a design matrix with $O(Ld)$ candidate parents and fits a spline-based GAM, yielding a per-variable cost $O(BT(Ld)^2)$; aggregated over all $d$ variables, this gives a worst-case pruning complexity of $O(BTL^2d^3)$.

Combining all components, a single EM iteration incurs $O(BTLd^2)+O(ILBTdp)+O(BTL^2d^3)$. In practice, for small to moderate (d), the kernel regression term $O(ILBTdp)$ dominates; the theoretical worst-case term $O(BTL^2d^3)$ becomes significant only when the number of variables grows large.

## A.11 Implement Details

This section outlines the implementation details of the ReTimeCausal framework used in our experiments. We describe the model initialization, regression module configuration, kernel choices, imputation strategies, optimization routines, and hyperparameter settings. These settings are consistent across all experimental conditions unless otherwise noted. The goal is to facilitate reproducibility and clarify the technical setup behind the reported results.

For baseline algorithms, we primarily follow the parameter settings recommended in the original papers or their official repositories (e.g., PCMCI[1], DYNOTEARS (Pamfil et al., 2020), Rhino[2], AERCA[3], TimeMixer[4], and CUTS+[5]). To ensure a fair comparison, we additionally perform hyperparameter tuning to identify the optimal settings for each baseline. The tuned hyperparameters are summarized in Table 12. All experiments were conducted on a single NVIDIA RTX 4090 GPU with 24GB memory, with each training run taking approximately 10–30 minutes depending on the dataset size.

Table 12: Hyperparameter Settings Used Across main Datasets and Methods

| Methods | Hyperparam | LR-10-10 | TANH-10-10 | TANH-20-20 | SIN-20-20 | CausalRivers | NetSim |
|---|---|---|---|---|---|---|---|
| PCMCI | $PC_\alpha$ | 0.05 | 0.05 | 0.05 | 0.05 | 0.05 | 0.05 |
| | CI Test | Parr | Parr | Parr | Parr | Parr | Parr |
| DYNOTEARS* | $\mathbf{W}$ lr | $10^{-3}$ | $10^{-3}$ | $10^{-3}$ | $10^{-3}$ | $10^{-3}$ | $10^{-3}$ |
| | $p$ | - | 400 | 400 | 400 | 400 | 400 |
| | $\sigma^2$ | - | 10 | 10 | 10 | 10 | 10 |
| | $\ell_1$ | - | $10^{-4}$ | $10^{-4}$ | $10^{-4}$ | $10^{-4}$ | $10^{-4}$ |
| CUTS+ | input_step | 1 | 2 | 2 | 2 | 2 | 3 |
| | n_groups | 4 | 4 | 8 | 8 | 4 | 2 |
| | model | multi_lstm | multi_lstm | multi_lstm | multi_lstm | multi_lstm | multi_lstm |
| | pred lr | $10^{-3} \to 10^{-4}$ | $10^{-3} \to 10^{-4}$ | $10^{-3} \to 10^{-4}$ | $10^{-3} \to 10^{-4}$ | $10^{-3} \to 10^{-4}$ | $10^{-3} \to 10^{-4}$ |
| | graph lr | $10^{-1} \to 10^{-2}$ | $10^{-1} \to 10^{-2}$ | $10^{-1} \to 10^{-2}$ | $10^{-1} \to 10^{-2}$ | $10^{-1} \to 10^{-2}$ | $10^{-2} \to 10^{-3}$ |
| | gru_layers | 3 | 3 | 3 | 3 | 3 | 3 |
| | lambda_s | $10^{-2} \to 10^{-3}$ | $10^{-2} \to 10^{-3}$ | $10^{-2} \to 10^{-3}$ | $10^{-2} \to 10^{-3}$ | $10^{-2} \to 10^{-3}$ | $10^{-2} \to 10^{-3}$ |
| | $\tau$ | $1 \to 0.1$ | $1 \to 0.1$ | $1 \to 0.1$ | $1 \to 0.1$ | $1 \to 0.1$ | $1 \to 0.1$ |
| Rhino | gumbel_temp | 0.25 | 0.25 | 0.25 | 0.25 | 0.25 | 0.25 |
| | lr | $10^{-3}$ | $10^{-3}$ | $10^{-3}$ | $10^{-3}$ | $10^{-3}$ | $10^{-3}$ |
| AERCA | window_size | 2 | 2 | 2 | 2 | 2 | 3 |
| | $\beta$ | 0.5 | 0.5 | 0.5 | 0.5 | 0.5 | 0.5 |
| | $\gamma_{encoder}$ | 0.1 | 0.15 | 0.15 | 0.15 | 0.15 | 0.15 |
| | $\lambda_{encoder}$ | 0.01 | 0.2 | 0.2 | 0.2 | 0.2 | 0.2 |
| | lr | $10^{-4}$ | $10^{-4}$ | $10^{-4}$ | $10^{-4}$ | $10^{-4}$ | $10^{-4}$ |
| | hidden layer | 2 | 4 | 4 | 4 | 4 | 4 |
| | layer size | 50 | 100 | 100 | 100 | 100 | 100 |
| ReTimeCausal | $f_i$ lr | $10^{-3}$ | $10^{-3}$ | $10^{-3}$ | $10^{-3}$ | $10^{-3}$ | $10^{-3}$ |
| | noise_scale | - | 0.2 | 0.2 | 0.2 | 0.2 | 0.2 |
| | $\mathbf{W}$ lr | - | $10^{-3}$ | $10^{-3}$ | $10^{-3}$ | $10^{-3}$ | $10^{-3}$ |
| | $p$ | - | 400 | 400 | 400 | 400 | 400 |
| | $\sigma^2$ | - | 10 | 10 | 10 | 10 | 10 |
| | $\lambda$ | $10^{-2}$ | $10^{-4}$ | $10^{-4}$ | $10^{-4}$ | $10^{-4}$ | $10^{-4}$ |

## A.12 Limitations

Despite its strong empirical performance and theoretical grounding, ReTimeCausal has several limitations that merit discussion. First, the method assumes that the missing data mechanism follows either the Missing Completely at Random (MCAR) or Missing At Random (MAR) assumptions. In practice, however, particularly in domains such as healthcare and finance, missingness is often Missing Not At Random (MNAR), where the probability of missing values depends on the unobserved data itself. ReTimeCausal does not explicitly model this setting, and applying it under MNAR may lead to biased estimates.

---

[1]https://github.com/jakobrunge/tigramite

[2]https://github.com/microsoft/causica

[3]https://github.com/hanxiao0607/AERCA

[4]https://github.com/kwuking/TimeMixer

[5]https://github.com/jarrycyx/UNN

Second, the current framework only models lagged causal relationships and explicitly disallows instantaneous effects—i.e., causal dependencies occurring within the same time step. While this assumption is valid in many real-world applications where causal effects unfold over time, it limits applicability in high-frequency or coarsely sampled settings, where instantaneous interactions may exist.

From a computational perspective, while the M-step of our EM algorithm is efficient due to the use of Random Fourier Feature approximations, the E-step becomes increasingly expensive as the number of variables, time lags, and missing entries grows. This bottleneck in the imputation step may hinder scalability to high-dimensional or long-horizon time series.

Another limitation lies in the stationarity assumption of the underlying causal mechanisms. ReTime-Causal assumes Consistency Throughout Time and fixed causal graphs, which precludes modeling dynamically evolving causal relationships—a direction that would be valuable for time-varying systems.

Finally, the empirical evaluation is currently limited to synthetic datasets and a subset of real-world benchmarks with moderate size and complexity. While results are promising, further large-scale validation—especially in domains like climate modeling, electronic health records, or financial transaction systems—would strengthen the generalizability and practical impact of the proposed approach.

