# OpenReview forum: "RETIMECAUSAL: A CONSISTENT EM FRAMEWORK FOR CAUSAL DISCOVERY IN IRREGULAR TIME SERIES"
_ICLR.cc/2026/Conference — ICLR 2026 Conference Desk Rejected Submission_

### Official Review · Reviewer_UPVv · 2025-10-27

**Soundness:** 3
**Presentation:** 2
**Contribution:** 2
**Rating:** 2
**Confidence:** 4

**Summary:**

This paper addresses a critical challenge in temporal causal discovery: the presence of missing values and inconsistent sampling frequencies in multivariate time series data. These data issues commonly lead to distortions that severely weaken the performance of subsequent causal inference methods. To overcome this limitation, the authors propose a novel, unified framework that tackles the problems of alignment and imputation concurrently with causal structure learning. The core of their solution is an Expectation-Maximization (EM) style method that allows the two traditionally separate processes to be mutually promoted, thereby improving the robustness and accuracy of the resulting causal graph.

**Strengths:**

1. The paper presents a well-motivated approach and proposes a framework that is both simple and effective.
2. The authors provide a compelling argument challenging the traditional two-step approach (imputation followed by causal discovery), emphasizing instead that these two processes can and should be mutually promoted. This insight forms the basis of their innovative approach.

**Weaknesses:**

1. The authors need to clarify the definitions of numerous mathematical symbols. For instance, the variable $k$ appears to denote the number of iteration. Overall, the mathematical notation and methodology presentation in the paper require significant improvement for clarity and rigor.
2. The empirical evaluation lacks sufficient breadth to fully validate the proposed framework's generalizability and robustness. Specifically, the authors should include comparisons against a wider range of competitive state-of-the-art methods and ideally use more diverse real-world datasets to demonstrate the method's effectiveness across different domains and data characteristics.

**Questions:**

1. (Page 3, Equation 1): Regarding the ANM framework employed, could the authors please clarify whether the proposed approach explicitly accounts for and models contemporaneous effects (instantaneous effects) between variables?

2. (Page 5, Equation 3): When dealing with consecutive missing values (as depicted in Figure 2), does the calculation rely solely on directly observed data, or does it utilize autoregressively imputed values from the previous timestep as input?

3. (Page 5, Equation 4): The paper needs to clarify the functional form of $f_i$. The authors state that black-box neural networks are not used; however, it remains ambiguous whether the inherent interpolation or modeling of $f_i$ is strictly linear, a form of low-order non-linearity, or another mechanism.

4. Clarification on the RKHS and dimensional handling is essential. While we infer the input space dimension to be $p$ (based on the $p \times p$ matrix $\mathbf{W}$), please confirm this and explain the conceptual role and effective dimensionality of the RKHS. Furthermore, a major inconsistency exists in the regularization output: if $\mathbf{W}$ is $p \times p$, and $\mathbf{G}$ is derived via LASSO, how is the dimension of $\mathbf{G}$ subsequently reduced or transformed from $p \times p$ to $d \times d$?

5. The selection of the crucial threshold parameter should be clarified. Please detail the methodology used to determine this parameter in the experiments, and provide recommended strategies or principles for practitioners seeking to tune it for real-world applications.

6. The current set of comparison methods is too limited. The authors should include a wider range of modern and relevant time series causal discovery baselines. Specifically, please evaluate powerful two-step processes, i.e., combining the most advanced imputation methods with established discovery techniques. Please find a list of relevant references provided below for your consideration.

7. The superior performance of PCMCI + TimeMixer in Table 3 requires a detailed analysis to clarify its mechanism. Does this superior performance primarily stem from the high quality of the TimeMixer imputation? If so, does this imply that combining TimeMixer with other temporal causal discovery methods would also achieve similar high performance? If the strength lies uniquely in the PCMCI method itself, the authors could provide a clear justification for why PCMCI is uniquely well-suited for the TimeMixer-imputed outputs in this missing data scenario.

8. The paper lacks an ablation study to substantiate the central claim of mutual promotion between imputation and discovery. Could the authors provide a visualization (a plot over iterations $k$) showing how both the causal recovery accuracy and the imputation quality evolve, demonstrating how they promote each other?

9. While the paper addresses inconsistent sampling, clarification is needed on how the alignment mechanism handles time distortions (e.g., issues typically addressed by dynamic time warping). How is this addressed, and how does the model differentiate between a true missing block and a time warp?

References:

a. Han et al. "Root Cause Analysis of Anomalies in Multivariate Time Series through Granger Causal Discovery." ICLR 2025.

b. Gong et al. "Rhino: Deep causal temporal relationship learning with history-dependent noise." ICLR 2023.

c. Gao et al. "IDYNO: Learning nonparametric DAGs from interventional dynamic data." ICML 2022.

d. Marcinkevičs et al. "Interpretable models for granger causality using self-explaining neural networks." ICLR 2021.

---

> ### Author Response · Authors · 2025-11-23
> **Part 1/3: Notation; More baseline and new dataset; instantaneous effects; Missing-Value imputation**
>
> > **(W1): Clarify the definitions of numerous mathematical symbols.**
>
> We thank the reviewer for pointing out the ambiguity in our mathematical notation. The variable $k$ indeed denotes the index of the current EM iteration. To avoid confusion with other uses of $k$, we have revised the notation in the updated version and replaced $k$ with $r$ in the original Equation (6).
>
> In addition, to improve clarity and rigor, we have added a new section **Appendix A.2 (Notations Used Throughout This Paper)**, which provides a table listing all key symbols used in the paper along with their definitions. We believe these revisions substantially enhance the readability and precision of the mathematical presentation.
>
> > **(W2 & Q6): The current set of comparison methods is too limited. The authors should include a wider range of modern and relevant time series causal discovery baselines and ideally use more diverse real-world datasets to demonstrate the method's effectiveness across different domains and data characteristics. **
>
> Thank you for raising this point. We agree that a broader empirical evaluation is important. To address this concern, we expanded both the baseline set and the set of real-world datasets.
>
> First, **we added two recent time-series causal modeling methods**: Rhino and AERCE. Rhino can model nonlinear instantaneous and lagged relations, but it is designed for fully observed sequences and does not address irregular sampling. AERCE is mainly built for anomaly analysis, but it includes a causal scoring module. Adding these two methods gives a more complete set of baselines. We re-ran all experiments with identical settings to ensure fairness.
>
> Second, **we added a real-world dataset**. We now include the PM2.5 air-quality dataset from CausalTime (see Table 10 in revised version), which has nonlinear behavior and strong temporal variation. Together with CausalRivers (real hydrological data), CausalTime medical records, and the biophysically realistic NetSim benchmark, our evaluation now covers environmental data, healthcare data, and physical systems. These datasets provide different forms of noise, irregularity, and temporal structure.
>
> For space reasons, we were not able to include all new tables directly in the rebuttal. We have added the full results to the revised version, and we kindly invite the reviewer to see Section 5, Appendix A.8, Appendix A.9, and Appendix A.11. Across these new datasets and baselines, ReTimeCausal remains stable and competitive, especially under high missingness and irregular sampling. We hope these updates address your concern.
>
> > **(Q1): Regarding the ANM framework employed, could the authors please clarify whether the proposed approach explicitly accounts for and models contemporaneous effects (instantaneous effects) between variables**
>
> As stated in the *Limitations* section (see line 1409 in original version or line 1556 in revised version), the current framework does not explicitly model contemporaneous (instantaneous) effects between variables.
>
> > **(Q2): When dealing with consecutive missing values (as depicted in Figure 2), does the calculation rely solely on directly observed data, or does it utilize autoregressively imputed values from the previous timestep as input?**
>
> Thank you for the question. When we encounter consecutive missing values, we use all available values from the previous $L$ timesteps, where $L$ is the maximum lag. This set includes both directly observed entries and imputed entries from earlier time points. In other words, the imputation is autoregressive along the time axis: once a value at time $t$ is imputed, it can be used as an input when imputing time $t+1$ within the same E-step. At the same time, we recompute the entire sequence of imputations at every EM iteration under the updated structural functions, so earlier errors can be corrected rather than propagating unchecked across iterations.

---

> ### Author Response · Authors · 2025-11-23
> **Part 2/3: Functional form; RKHS and dimensional handling; Selection strategy of threshold parameter**
>
> > **(Q3): The paper needs to clarify the functional form of $f_i$. The authors state that black-box neural networks are not used; however, it remains ambiguous whether the inherent interpolation or modeling of $f_i$ is strictly linear, a form of low-order non-linearity, or another mechanism.**
>
> Thank you for the question. We clarify the form of $f_i$. As described in the paper, each structural function $f_i$ is implemented as a small 3-layer MLP with a fixed hidden width. This MLP is used to evaluate $f_i(\text{Pa}_i)$ during imputation in the E-step. The nonlinearity of $f_i$ therefore comes from this neural network, and it is not restricted to linear or low-order polynomial forms. Unlike end-to-end neural models, ReTimeCausal decomposes learning into explicit, interpretable EM steps—structure-aware imputation (E-step) and sparse causal update (M-step). These steps make the structure learning process observable and traceable. Thus, the interpretability of ReTimeCausal lies in its transparent algorithmic process rather than in restricting $f_i$ to simple functional forms.
>
> > **(Q4): Clarification on the RKHS and dimensional handling is essential. While we infer the input space dimension to be $p$ (based on the $p\times p$ matrix $\mathbf{W}$), please confirm this and explain the conceptual role and effective dimensionality of the RKHS. Furthermore, a major inconsistency exists in the regularization output: if $\mathbf{W}$ is $p\times p$, and $\mathbf{G}$ is derived via LASSO, how is the dimension of $\mathbf{G}$ subsequently reduced or transformed from $p\times p$ to $d\times d$?**
>
> We thank the reviewer for the careful observation. The RKHS is not used to define the causal graph; rather, it provides an intermediate function space in which the nonlinear structural functions of the ANM become linear and thus amenable to sparse regression. The original input dimension is fixed to **d**, while each variable is mapped to a **p-dimensional random-feature representation**, where $p$ controls the approximation fidelity of the underlying infinite-dimensional RKHS. Larger $p$ increases expressiveness but does not change the fact that causal edges must ultimately be expressed in the **d-dimensional input space**.
>
> Within this kernel feature space, we learn lag-specific weight matrices $W_{\text{high}}^{(\tau)}$. The matrix at *line 266* indeed contains a typographical error: the correct shape is $W_{\text{high}}^{(\tau)} \in \mathbb{R}^{p \times d}$, not $p \times p$. After regression, we project these weights back to the input space using the fixed feature map $\Phi \in \mathbb{R}^{d \times p}$, ensuring dimensional consistency and interpretability. The corrected form of Eq. (6) now reads:$W^{(\tau)}\_{j,i}=\sum_{r=1}^{p}\Phi_{j,r}, W^{(\tau)}_{\text{high},r,i}$, which naturally yields a $d \times d$ adjacency matrix after thresholding.
>
> > **(Q5): The selection of the crucial threshold parameter should be clarified. Please detail the methodology used to determine this parameter in the experiments, and provide recommended strategies or principles for practitioners seeking to tune it for real-world applications.**
>
> We appreciate the reviewer bringing up this question. To clarify the selection of crucial threshold parameter($\alpha$, $\beta$, $\gamma$), we conducted a systematic sensitivity analysis and included the results in Appendix A.9.7. Every hyperparameter in this study was changed separately over a wide range while the others remained constant. The observed trends were very consistent, and the experiments were conducted under 60% missingness on a representative nonlinear configuration (TANH–laplace–20–40–1).
>
> **(1) Smoothing Coefficient  $\alpha\in \{0.7, 0.8, 0.9\}$**
>
> | value | SHD  | F1      |
> | ----- | ---- | ------- |
> | 0.7   | 4    | 0.952 |
> | 0.8   | 5    | 0.938 |
> | 0.9   | 3    | 0.964 |
>
> **(2) Pruning threshold $\beta \in \{0.01, 0.05, 0.10\}$**
>
> | value | SHD  | F1      |
> | ----- | ---- | ------- |
> | 0.01  | 4    | 0.951 |
> | 0.05  | 5    | 0.938 |
> | 0.10  | 7    | 0.920 |
>
> **(3) Graph binarization threshold $\gamma\in \{0.05, 0.15, 0.25\}$**
>
> | value | SHD  | F1      |
> | ----- | ---- | ------- |
> | 0.05  | 5    | 0.938 |
> | 0.15  | 9    | 0.892 |
> | 0.25  | 9    | 0.883 |
>
> We suggest the following principled selection approach in light of these findings:
>
> - Selecting $\alpha$ in the stable range [0.8, 0.9] for the smoothing coefficient $\alpha$ consistently produces smooth iterative updates without unduly suppressing new structural evidence.
>
> - CAM pruning threshold $\beta$: set $\beta$ to a stringent but moderate value, like 0.01–0.05. This prevents the permissive behavior seen at $\beta=0.10$ while enforcing significant statistical pruning.
> - Binarization threshold $\gamma$: to preserve weak-but-true coefficients for CAM's significance-based filtering, use a small $\gamma$ around 0.05 (or within 0.05–0.15).

---

> ### Author Response · Authors · 2025-11-23
> **Part 3/3: Mechanism of PCMCI + TimeMixer performance; Imputation and discovery;  Time distortions**
>
> > **(Q7): The superior performance of PCMCI + TimeMixer in Table 3 requires a detailed analysis to clarify its mechanism.**
>
> Thank you for the question. We have added an extended analysis after Table 3 to clarify this behavior. The strong score of PCMCI+TimeMixer appears only in the low-missingness setting of this dataset and does not generalize across conditions. As shown in Table 3, PCMCI+TimeMixer performs reasonably well at 20% missingness (F1 = 0.414), but this pattern disappears once the missing rate increases to 60%. In contrast, ReTimeCausal remains the strongest method across both missing rates, achieving the highest F1 scores and the lowest SID under high missingness.
>
> This indicates that the apparent advantage of PCMCI+TimeMixer occurs only under low missingness. When the missing rate is small, TimeMixer produces smooth interpolations that work well with PCMCI’s conditional-independence tests. However, when the missing rate increases, the imputation becomes less reliable, and the additional noise introduced by interpolation weakens PCMCI’s tests. As a result, PCMCI+TimeMixer no longer maintains the same level of performance in the high-missingness setting.
>
> Overall, the behavior in Table 3 shows up only in this dataset and when the missing rate is low, rather than reflecting a general advantage of this pipeline. We have incorporated this clarification into Section 5.3 of the revised manuscript.
>
> > **(Q8): The paper lacks an ablation study to substantiate the central claim of mutual promotion between imputation and discovery.**
>
> We thank the reviewer for the valuable feedback. To illustrate how both causal recovery accuracy and imputation quality evolve during training, we conducted additional experiments (5 independent runs under the LR-gaussian-20-20-1 setting) and added a new section in Appendix A.9.6 for this ablation study.
> Figure 4 (new in the revised paper) presents the iteration-wise evolution curves averaged over these 5 experiments, with shaded areas indicating the standard error. The results consistently demonstrate that imputation and discovery reinforce each other across different random initializations.
> We believe these revised findings further strengthen our claims and improve the overall quality of the paper.
>
> > **(Q9): While the paper addresses inconsistent sampling, clarification is needed on how the alignment mechanism handles time distortions (e.g., issues typically addressed by dynamic time warping). How is this addressed, and how does the model differentiate between a true missing block and a time warp?**
>
> Thank you for the question. ReTimeCausal focuses on irregular sampling, where different variables are observed at different timestamps. When we place all variables on a shared time grid, this produces missing entries. Our model imputes these missing values, and this is the only form of temporal misalignment we address. We do not model nonlinear time distortions or time warping, as done in DTW. The time axis itself is assumed to be reliable.
>
> Regarding the distinction between missing blocks and time warps: in our problem setting, all unobserved segments come from asynchronous sampling rather than temporal deformation. The alignment step simply maps observations to the nearest grid point and leaves the timestamps unchanged. This design matches the domains we study, where uneven sampling is common but actual time warping is not part of the data-generating process.

---

> ### Author Response · Authors · 2025-11-23
>
> If you have any further questions or concern, please feel free to let us know, we will do our best to address them.

---

> ### Comment · Area_Chair_Xgcw · 2025-11-27
> **Please reply to the authors' rebuttal**
>
> Dear Reviewer,
>
> The authors have provided their rebuttal. Please reply to it before the rebuttal period ends. Thanks!
>
> Best regards,
>
> AC

---

### Official Review · Reviewer_CgG6 · 2025-11-01

**Soundness:** 3
**Presentation:** 3
**Contribution:** 3
**Rating:** 8
**Confidence:** 4

**Summary:**

The paper addresses causal discovery when multivariate time series are irregularly sampled and contain substantial missingness. Standard interpolation or alignment steps often distort causal order and create spurious dependencies.
The authors propose ReTimeCausal, an Expectation–Maximization (EM) style framework that alternates between

1. a structure-aware imputation step that estimates missing values based on conditional expectations under additive-noise models, and

2. a structure-learning step that performs kernelized sparse regression with projection to input space and thresholding to infer lagged causal graphs.


A noise-aware imputation mechanism injects residual noise to preserve independence assumptions used for pruning. Theoretical analysis (Proposition 1) establishes structural consistency under standard assumptions — MCAR/MAR missingness, finite-order Markovity, causal sufficiency, and faithfulness — given appropriate smoothing and threshold schedules.
Experiments on synthetic linear and nonlinear systems and the CausalRivers dataset show that ReTimeCausal maintains high F1-scores even with 60–80 % missingness and outperforms baselines such as PCMCI, DYNOTEARS, and CUTS+ (with TimeMixer preprocessing). On CausalRivers, it achieves the best reported F1 = 0.463 (MR = 0.2) and 0.414 (MR = 0.6).

**Strengths:**

- Principled EM alternation: directly enforces consistency between imputation and structure learning.
- Kernelized sparse regression: captures nonlinear lagged effects and projects to interpretable input-space graphs.
- Noise-aware imputation: preserves independence assumptions for accurate CAM-style pruning.
- Theoretical support: provides asymptotic structural consistency (Proposition 1) under clear conditions.
- Empirical robustness: strong performance across missingness levels; best F1 on CausalRivers with competitive SID/SHD.

**Weaknesses:**

- Assumption scope: relies on MCAR/MAR and causal sufficiency; MNAR and latent confounding remain open.
- Hyperparameter sensitivity: thresholds (γ, β) and smoothing (α) impact performance but lack systematic tuning guidance.
- Computational profile: runtime and memory costs for kernel features plus EM iterations are not detailed.
- Evaluation scale: real-world validation limited to a 10-node CausalRivers subgraph; larger datasets would improve external validity.

**Questions:**

1. How sensitive are the results to thresholds (γ, β) and smoothing (α)? Any principled selection strategy?
2. Can you provide empirical convergence diagnostics (E/M objective traces or iteration counts)?
3. How does the method behave under mild MNAR violations where missingness depends weakly on unobserved values?
4. What are the main computational bottlenecks, and can kernel features be approximated (e.g., random features)?
5. How does CAM-style pruning compare with simpler coefficient-magnitude pruning in nonlinear regimes?

---

> ### Author Response · Authors · 2025-11-23
> **Part 1/3: MNAR; Hyperparameter sensitivity**
>
> Part 1/3: MNAR; Hyperparameter sensitivity
>
> > **(W1 & Q3): Relies on MCAR/MAR and causal sufficiency; MNAR and latent confounding remain open. How does the method behave under mild MNAR violations where missingness depends weakly on unobserved values?**
>
> We thank the reviewer for raising this important point. Our theoretical guarantees rely on ignorable missingness (MCAR/MAR) and causal sufficiency. We agree that it is helpful to check how the method behaves when the missingness process has a weak dependence on the unobserved values.
>
> To address the reviewer’s concern, we added a new experiment using a *light self-masked MNAR mechanism*. Specifically, we modify the missingness process from an ignorable MAR baseline to a *mild MNAR* mechanism in which missingness depends slightly on the (unobserved) true value:$
> \text{logit}\, p_{t,i}= \log\frac{\mathrm{MR}}{1-\mathrm{MR}}+\rho \cdot \tilde X_{t,i},
> $
> where $p_{t,i}$ is the probability of missingness at position $(t,i)$, $\tilde X_{t,i}$ is the standardized underlying value, $\rho \in \{1, 3, 5\}$ controls the strength of MNAR dependence, $\rho=0$ corresponds to the MAR case used in our theoretical analysis.
>
> | value | SHD  | F1      |
> | ----- | ---- | ------- |
> | 1     | 2    | 0.952 |
> | 3     | 8    | 0.750 |
> | 5     | 13   | 0.552 |
>
> From the above results on the TANH nonlinear synthetic dataset (20 nodes, 20 edges, 1 lag), we observe that the method remains stable when the MNAR dependence is mile($\rho=1$). As the MNAR signal becomes stronger, the performance decreases, which is expected because the missing entries depend on values that are not observed. This result is consistent with the theoretical scope of our method, since our EM procedure assumes ignorable missingness and does not model the missingness mechanism itself.
>
> > **(W2 & Q1): Hyperparameter sensitivity: thresholds (γ, β) and smoothing (α) impact performance but lack systematic tuning guidance. How sensitive are the results to thresholds (γ, β) and smoothing (α)? Any principled selection strategy?**
>
> We appreciate the reviewer’s feedback. We agree that hyperparameter robustness is crucial for practical deployment. To address this concern, we conducted three sets of hyperparameter sensitivity experiments. Specifically, we vary each hyperparameter independently across a broad range while keeping all others fixed, evaluated on a representative nonlinear configuration (TANH–laplace–20–40–1) under 60% missingness. The results are summarized below.
>
> **(1) Smoothing Coefficient  $\alpha\in \{0.7, 0.8, 0.9\}$**
>
> | value | SHD  | F1      |
> | ----- | ---- | ------- |
> | 0.7   | 4    | 0.952 |
> | 0.8   | 5    | 0.938 |
> | 0.9   | 3    | 0.964 |
>
> **(2) Pruning threshold $\beta \in \{0.01, 0.05, 0.10\}$**
>
> | value | SHD  | F1      |
> | ----- | ---- | ------- |
> | 0.01  | 4    | 0.951 |
> | 0.05  | 5    | 0.938 |
> | 0.10  | 7    | 0.920 |
>
> **(3) Graph binarization threshold $\gamma\in \{0.05, 0.15, 0.25\}$**
>
> | value | SHD  | F1      |
> | ----- | ---- | ------- |
> | 0.05  | 5    | 0.938 |
> | 0.15  | 9    | 0.891 |
> | 0.25  | 9    | 0.883 |
>
> The experimental results show that the (i)Smoothing Coefficient $\alpha$ maintains stable performance in the range of 0.7-0.9, and the F1 value fluctuates only within a narrow range (0.938–0.964). (ii)A moderate pruning threshold ($\beta=0.01$ ) yields robust performance, while an excessively high threshold ($\beta=0.10$) prevents CAM from trimming more false edges, leading to performance degradation. (iii)A small $\gamma$ (0.05–0.15) works best because it avoids prematurely discarding weak-but-true causal coefficients. Keeping more candidate parents at this stage allows CAM to perform pruning based on conditional independence, rather than losing information due to overly aggressive thresholding.
>
> Based on these observations, we recommend the following principled selection strategy:
>
> - Smoothing coefficient α: choose $\alpha$ in the stable range [0.8, 0.9], which consistently yields smooth iterative updates without over-suppressing new structural evidence.
> - CAM pruning threshold $\beta$: set β=$\beta$ to a strict but not overly aggressive value such as 0.01–0.05. This enforces meaningful statistical pruning while avoiding the permissive behavior observed at $\beta=0.10$.
> - Binarization threshold $\gamma$: use a small $\gamma$ around 0.05 (or within 0.05–0.15) to retain weak-but-true coefficients for CAM’s significance-based filtering.
>
> We have added all these finding in Appendix A.9.7 of the revised version.

---

> ### Author Response · Authors · 2025-11-23
> **Part 2/3: Runtime and memory costs; Time complexity**
>
> > **(W3): Runtime and memory costs for kernel features plus EM iterations are not detailed.**
>
> Thank you for the helpful comment. We agree that a clear runtime and memory profile is important. We provide a detailed analysis in Appendix A.10. Here we summarize the main results.
>
> Our method has three sources of cost: (1) filling missing values in E-steps, (2) kernelized sparse regression in M-steps, and (3) CAM-style pruning. Let the batch size be $B$, sequence length $T$, number of variables $d$, lag order $L$, random feature dimension $p$, optimizer iterations $I$, and missing rate $\rho$.
>
> **(1) Runtime.**
>  The total time complexity is$O(BTLd^2) + O(ILBTdp) + O(BTL^2 d^3).$
>
> The E-step cost is $O(BTLd^2)$. Each missing entry needs to retrieve its parent values, and this takes $O(Ld)$ time. Accumulating over $\rho B T d$ missing entries gives the above term.
>
> The main M-step cost comes from kernelized regression and random Fourier features. Constructing features and updating model weights takes $O(IL B T d p)$. This cost scales linearly with $p$ and does not depend on the Gram matrix.
>
> CAM pruning adds another term. For each variable, we build a design matrix with $O(L d)$ parent candidates and fit a spline-based additive model. This leads to $O(B T (L d)^2)$ per variable, and $O(B T L^2 d^3)$ in total.
>
> **(2) Memory.**
>  The peak memory use is$O(BTLp + BTd^2 + Ldh).$
>
> Kernel features use $BTLp$ floats. The filled data and gradients use $2BTd$ floats. The MLP parameters use $O(Ldh)$. CAM pruning allocates a temporary matrix of size $BTd^2$. These costs grow at most quadratically in $d$ and remain moderate for the graph sizes used in our experiments.
>
> We hope this profile addresses the reviewer’s concern.
>
> > **(W4): Real-world validation limited to a 10-node CausalRivers subgraph; larger datasets would improve external validity.**
>
> Thank you very much for pointing this out. We agree that evaluating on larger real-world datasets can further strengthen external validity. To address your cincern, we have conducted an additional set of experiments on **larger CausalRivers subgraphs**.
>
> Specifically, we extend our evaluation from the original 10-node subgraph to **30-node and 50-node CausalRivers subgraphs**, which naturally contain more river stations, higher spatial heterogeneity, and stronger temporal variability. This expansion significantly increases the scale and complexity of the real-world benchmark, thereby providing a more rigorous test of external validity while preserving the intrinsic high-noise and nonstationary characteristics of CausalRivers.
>
> The new results are shown below:
>
> | value  | SHD  | F1    |
> | ------ | ---- | ----- |
> | $d=10$ | 36   | 0.463 |
> | $d=30$ | 65   | 0.356 |
> | $d=50$ | 119  | 0.320 |
>
> The F1 score becomes lower as the graph grows, which is expected because the task becomes more complex. The method still keeps stable behavior across all three sizes and does not show signs of failure. These results suggest that ReTimeCausal remains effective on larger and more difficult real-world subgraphs.

---

> ### Author Response · Authors · 2025-11-23
> **Part 3/3: Empirical convergence diagnostics; Computational bottlenecks; Pruning**
>
> > **(Q2): Can you provide empirical convergence diagnostics (E/M objective traces or iteration counts)?**
>
> Thank you for the helpful suggestion. We agree that empirical convergence diagnostics are important. To dispel your concerns, we have added a new analysis in Appendix A.9.6. In ReTimeCausal, both the E-step and M-step give task-level signals that we can monitor during training. We therefore report the iteration-wise curves of two natural and stable indicators(MSE and F1).
>
> For the E-step, we track the imputation MSE on the completed sequences. This value shows how well the model fits the observed and missing data. As training goes on, the MSE keeps decreasing but its rate of change becomes small and smooth across runs.
>
> For the M-step, we track the F1 score of the recovered binary causal graph. The M-step outputs this graph directly. The F1 score increases fast in the early stage and then reaches a stable level.
>
> Figure 4 (new in the revised version)  shows both curves averaged over 5 runs. The F1 curve becomes stable first, while the MSE curve continues to decrease at a slower rate. This pattern suggests that structure learning reaches a steady point earlier, and imputation continues to refine the signal with small updates. Together, these curves provide a clear and practical view of empirical convergence in our setting.
>
> > **(Q4): What are the main computational bottlenecks, and can kernel features be approximated (e.g., random features)?**
>
> Thank you for the question. As shown in our response to W3, the total time complexity of our method is
> $$
> O(BTLd^2) + O(ILBTdp) + O(BTL^2d^3).
> $$
> When the graph is small, the main bottleneck comes from the kernel regression in the M-step. When the graph becomes larger, the CAM pruning step becomes the dominant cost.
>
> Regarding kernel efficiency: a full kernel implementation requires constructing an $N \times N$ Gram matrix with cost $O(N^2)$, where $N = B \times T$. To avoid this cost, we use Random Fourier Features (RFF) to approximate the kernel. This reduces the complexity to $O(Np)$, where $p$ is the number of random features. Yes — kernel features can be approximated, and we already use RFF to make the nonlinear regression scalable.
>
> > **(Q5): How does CAM-style pruning compare with simpler coefficient-magnitude pruning in nonlinear regimes?**
>
> Thank you for the helpful question. In nonlinear settings, the kernel model learns high-dimensional weights $\mathbf{W}_{\text{high}}$. These weights are then projected back to an input-level coefficient matrix $\mathbf{W}$. This projection mixes many nonlinear basis functions, so the size of each input-level coefficient does not reliably reflect the true effect of each parent variable. A real parent may get a small projected value, and a non-parent may get a large one. As a result, magnitude-based pruning is likely to result in more false negatives and false positives.
>
> CAM-style pruning works in a different way. It tests whether adding a variable gives a significant improvement to the structural function $f_i(\cdot)$. It does not depend on the projected coefficient size. This test can capture nonlinear but genuine contributions, and it can remove effects that come only from kernel feature mixing.

---

> ### Author Response · Authors · 2025-11-23
>
> If you have any questions or concerns, please feel free to let us know — we’d be happy to clarify, and we look forward to your response.

---

> > ### Comment · Reviewer_CgG6 · 2025-11-25
> >
> > Thanks for addressing my concerns! I will keep my initial score (8, accept).

---

> > > ### Author Response · Authors · 2025-11-27
> > >
> > > Thank you very much for taking the time to review our additional experiments and discussions. We are deeply grateful for your recognition of our work!

---

> ### Comment · Area_Chair_Xgcw · 2025-11-27
> **Please reply to the authors' rebuttal**
>
> Dear Reviewer,
>
> The authors have provided their rebuttal. Please reply to it before the rebuttal period ends. Thanks!
>
> Best regards,
>
> AC

---

### Official Review · Reviewer_eYxy · 2025-11-01

**Soundness:** 3
**Presentation:** 3
**Contribution:** 3
**Rating:** 6
**Confidence:** 3

**Summary:**

This paper introduces ReTimeCausal, an EM-based framework for causal discovery in multivariate time series with irregular sampling and high data missingness. By alternating between smart imputation and causal graph estimation, the method aims to ensure mutual consistency between the imputed values and the learned causal structure. The authors provide theoretical guarantees of consistency for structure recovery under a set of explicit assumptions. Experiments on synthetic and real-world datasets demonstrate improved performance over classical and recent baselines for both structural accuracy and robustness.

**Strengths:**

1. The EM-based approach is an intuitive solution to the often glossed over problem of missing data in prior work.
2. The theoretical results on structure consistency are justified and rigorous.
3. The kernelized sparse reg approach allows for nonlinear dynamics as well as linear.
4. Good recent baselines in experiments.

**Weaknesses:**

1. Although multiple real-world datasets are tested, the primary evaluation still leans heavily on synthetic data with relatively small- to medium-scale graphs (10–50 variables). The main real-world benchmark (CausalRivers) comprises a 10-node subgraph, and other datasets (NetSim, CausalTime) are also moderate in scale. Absence of large-scale, high-noise, and highly nonstationary real-world benchmarks undercuts the claimed generalizability.
2. Theoretical results hinge on a strong modeling assumption: MCAR/MAR missingness (Assumption 3.2). Many domains of practical interest, especially healthcare and finance, often involve MNAR data, latent confounding, or nonstationary relationships. While the authors acknowledge this (Appendix A.5, A.10), the framework offers no empirical or algorithmic pathway for relaxing these beyond a brief mention as future work.

### Minor comments:
3. You mention Assumption 3.1 in lines 145 and 357 but there is no such assumption in the text.

**Questions:**

1. Can the EM pipeline be extended to non-ignorable missingness (e.g., selection/Heckman-style or propensity-aware E-steps)?
2. What is the time complexity of ReTimeCausal?
3. How robust is the approach to severe nonstationarity and rapidly shifting lag structures?

---

> ### Author Response · Authors · 2025-11-23
> **Part 1/2: larger scale experiment; MNAR**
>
> > **(W1&Q3): Absence of large-scale, high-noise, and highly nonstationary real-world benchmarks undercuts the claimed generalizability. How robust is the approach to severe nonstationarity and rapidly shifting lag structures?**
>
> We thank the reviewer's feedback. CausalRivers is designed to capture strong noise and clear temporal nonstationarity in real river systems. The dataset also shows time-varying effective lags, because flow speed, rainfall patterns, and tributary inputs change across seasons. These changes shift the dominant lag inside the valid lag window. This setting gives us a natural way to test robustness under both severe nonstationarity and lag variation.
>
> To address the concern about larger and more complex systems, we run additional experiments on 30-node and 50-node subgraphs. These subgraphs keep the original nonstationary dynamics and the time-varying lag behavior of CausalRivers, and they increase the number of interacting variables. This setup makes the task harder and provides a stronger test of generalizability.
>
> ReTimeCausal remains stable under these settings:
>
> | value  | SHD  | F1    |
> | ------ | ---- | ----- |
> | $d=10$ | 36   | 0.463 |
> | $d=30$ | 65   | 0.356 |
> | $d=50$ | 119  | 0.320 |
>
> As we can see, the F1 score decreases when moving from 10 to 50 variables. While performance naturally decreases as the graph becomes larger and more nonstationary, ReTimeCausal still maintains reasonable and stable accuracy in these challenging regimes, demonstrating its robustness under high-noise and large-scale real-world conditions.
>
> > **(W2 & Q1): Theoretical results hinge on a strong modeling assumption: MCAR/MAR missingness (Assumption 3.2). The framework offers no empirical or algorithmic pathway for relaxing these beyond a brief mention as future work. Can the EM pipeline be extended to non-ignorable missingness (e.g., selection/Heckman-style or propensity-aware E-steps)?**
>
> We thank the reviewer for raising this important point. The current framework assumes an ignorable missingness mechanism (MCAR/MAR). Extending the method to non-ignorable missingness requires modeling the missingness process together with the data-generating process. Without this joint model, the E-step becomes non-identifiable under MNAR, because the missing entries depend directly on their unobserved values.
>
> Specifically, the EM pipeline can be extended through a selection-model formulation, a Heckman-style correction, or a propensity-aware E-step. All of these require an explicit model for $P(R \mid \mathbf{X}_m, \mathbf{X}_o)$, and they introduce additional parameters and identifiability challenges. Designing such a joint model is therefore non-trivial and lies beyond the current scope, but it provides a clear and meaningful direction for future work.
>
> To better understand the behavior of the method beyond MAR, we also performed a small check under a mild self-masked MNAR mechanism (the results are shown in our response to Reviewer CgG6, W1&Q3). The method remains stable when the MNAR dependence is weak, but it cannot correct strong MNAR bias without a model for the missingness process. This observation is consistent with the theoretical limitations described above.

---

> ### Author Response · Authors · 2025-11-23
> **Part 2/2: Typographical errors; Time complexity**
>
> > **(W3):  Assumption 3.1 in lines 145 and 357 but there is no such assumption in the text.**
>
> Thank you very much for your careful reading. We confirm that the occurrences of “Assumption 3.1” at lines 145 and 357 were unintended numbering mistakes.
>
> - At line 145, the correct reference should be “As formalized in Assumption 3.2”, referring to the ignorable missingness assumption.
> - At line 357, the correct statement should be “… satisfies Assumptions 3.2–3.6.”
>
> We have corrected both typographical errors in the revised version of the paper. We sincerely appreciate the reviewer’s careful attention to detail and for pointing out these mistakes.
>
> > **(Q2):  The time complexity of ReTimeCausal.**
>
> Thank you for the reviewer’s question. We provide a consolidated complexity analysis for both steps of ReTimeCausal. The total complexity is $O(BTLd^2) + O(ILBTdp) + O(BTL^2d^3)$.
>
> Here, $B$ is the batch size, $T$ is the time series length, $d$ is the number of variables, $L$ is the maximum lag, $p$ is the feature dimension, and $I$ is the number of optimizer steps. The brief analysis is as follows.
>
> **E-step**
>
> For each missing entry, retrieving its lagged parent values costs $O(Ld)$, and evaluating the structural function $f_i(\cdot)$ costs $O(C_f)$. Since each $f_i$ is a 3-layer MLP with constant hidden widths, we have $O(C_f)\approx O(Ld)$. With $M=\rho BTd$ missing entries, the E-step complexity is $O\big(M(Ld + C_f)\big)=O(\rho BTLd^2)=O(BTLd^2)$.
>
> **M-step**
>
> Updating all structural functions uses the valid samples and costs $O(BTLd^2)$.
>  Kernel regression uses random Fourier features and has cost $O(ILBTdp)$.
>  The CAM pruning step tests $O(Ld)$ candidates for each variable and has worst-case cost $O(BTL^2 d^3)$.
>
> In practice, for small–moderate $d$, the kernel regression term $O(ILBTdp)$ dominates, while the theoretical worst-case complexity $O(BTL^2d^3)$ becomes significant only when the number of variables grows large.
>
> We have added detailed analysis to Appendix A.10 in the revised version.

---

> ### Author Response · Authors · 2025-11-23
>
> If anything is unclear, just let us know — we’d be more than happy to provide further explanation or additional details.

---

> > ### Comment · Reviewer_eYxy · 2025-11-27
> >
> > I thank the authors for their detailed response and the additional effort put into running new experiments on larger graphs and providing the complexity analysis.

---

> ### Comment · Area_Chair_Xgcw · 2025-11-27
> **Please reply to the authors' rebuttal**
>
> Dear Reviewer,
>
> The authors have provided their rebuttal. Please reply to it before the rebuttal period ends. Thanks!
>
> Best regards,
>
> AC

---

### Author Response · Authors · 2025-11-29
**Summary of Revisions and Responses**

Dear ICLR 2026 AC, SAC, and PC,

Thank you very much for your time and effort in overseeing the review process. During the rebuttal stage, we carefully revised the manuscript to address all major technical and empirical concerns raised by the reviewers. We are grateful that the reviewers eYxy and CgG6 have kindly confirmed that their main concerns have been satisfactorily addressed and maintained their positive score.

The key revisions corresponding to the common issues raised by multiple reviewers are summarized below.

- **(Reviewer eYxy, CgG6)** For the concern regarding **limited real-world scale and external validity**, we added new large-scale real-world experiments on the 30-node and 50-node CausalRivers subgraphs to strengthen the evaluation of scalability and external validity;
- **(Reviewer eYxy, CgG6)** For the concern regarding the **MCAR/MAR assumption and MNAR applicability**, we expanded the theoretical discussion and added experiments under weak MNAR conditions, clarifying both the theoretical scope and empirical behavior;
- **(Reviewer eYxy, CgG6)** For the concern regarding **computational complexity**, we added a full computational complexity analysis in Appendix A.10, detailing the dominant computational components;
- **(Reviewer CgG6, UPVv)** For the concern regarding **hyperparameter sensitivity**, we performed systematic sensitivity analyses for α, β, and γ, and provided principled tuning guidelines in Appendix A.9.7;
- **(Reviewer CgG6, UPVv)** For the concerns regarding both **convergence diagnostics and the mutual promotion between imputation and causal discovery**, we added iteration curves (imputation MSE and discovery F1) in Appendix A.9.6. These curves jointly address the two issues by showing clear convergence behavior and by illustrating how the imputation and structure-learning components reinforce each other throughout the iterative process;

In addition to these shared concerns, we also addressed the specific points raised by individual reviewers as follows.

- **(Reviewer eYxy)** The incorrect references to “Assumption 3.1” have been corrected to “Assumption 3.2” in the revised manuscript (lines 145 and 364);
- **(Reviewer CgG6)** For the question on **pruning in nonlinear regimes**, we clarified why CAM-style pruning is more reliable than magnitude-based pruning;
- **(Reviewer UPVv)** For the detailed technical concerns, we made several clarifications and additions. Specifically, we (i) **added a complete notation table** (Appendix A.2), (ii) **incorporated two additional baselines (Rhino and AERCA) and added the PM2.5 dataset** from CausalTime (Section 5; Appendix A.8, A.9, A.11), and (iii) **clarified several methodological points**, including instantaneous effects, the use of imputed values, the functional form of $f_i$, RKHS dimensionality and projection (Section 4.2), the performance of PCMCI+TimeMixer (Section 5.3), and the distinction between missing blocks and temporal distortions;

Although Reviewer UPVv has not provided a follow-up response, we have carefully addressed all of the concerns in the rebuttal and incorporated the corresponding revisions into the manuscript. Given that the other two reviewers have confirmed that their concerns have been resolved and have maintained positive scores, we believe that the clarifications and updates in the manuscript help further address the concerns raised by Reviewer UPVv as well.

We sincerely appreciate your time and effort invested in improving the quality of our submission and the conference as a whole, especially given the unexpected challenges related to anonymity issues this year.

Best regards,

The authors of Paper 4401

---

### Note · Program_Chairs · 2026-01-17
**Submission Desk Rejected by Program Chairs**

The following references in this submission do not refer to real documents and/or have major errors in bibliographic information:

 Y. Xu et al. Predictive modeling of biomedical temporal data in healthcare: challenges
and solutions. Journal of Biomedical Informatics, 145:104379, 2024. doi: 10.1016/j.jbi.
2024.104379. URL https://www.sciencedirect.com/science/article/pii/
S1532046424000796.